# Constraining the budget of atmospheric carbonyl sulfide using a 3-D chemical transport model

**Michael P. Cartwright**[1,2], **Richard J. Pope**[3,4], **Jeremy J. Harrison**[1,2], **Martyn P. Chipperfield**[3,4], **Chris Wilson**[3,4], **Wuhu Feng**[3,5], **David P. Moore**[1,2], **and Parvadha Suntharalingam**[6]

[1]School of Physics and Astronomy, Space Park Leicester, University of Leicester, Leicester, UK
[2]National Centre for Earth Observation, Space Park Leicester, University of Leicester, Leicester, UK
[3]School of Earth and Environment, University of Leeds, Leeds, UK
[4]National Centre for Earth Observation, University of Leeds, Leeds, UK
[5]National Centre for Atmospheric Science, University of Leeds, Leeds, UK
[6]School of Environmental Sciences, University of East Anglia, Norwich, UK

**Correspondence:** Michael P. Cartwright (mpc24@leicester.ac.uk)

**Abstract.** Carbonyl sulfide (OCS) has emerged as a valuable proxy for photosynthetic uptake of carbon dioxide ($CO_2$) and is known to be important in the formation of aerosols in the stratosphere. However, uncertainties in the global OCS budget remain large. This is mainly due to the following three flux terms: vegetation uptake, soil uptake and oceanic emissions. Bottom-up estimates do not yield a closed budget, which is thought to be due to tropical emissions of OCS that are not accounted for. Here we present a simulation of atmospheric OCS over the period 2004–2018 using the TOMCAT 3-D chemical transport model that is aimed at better constraining some terms in the OCS budget. Vegetative uptake of OCS is estimated by scaling gross primary productivity (GPP) output from the Joint UK Land Environment Simulator (JULES) using the leaf relative uptake (LRU) approach. The remaining surface budget terms are taken from available literature flux inventories and adequately scaled to bring the budget into balance.

The model is compared with limb-sounding satellite observations made by the Atmospheric Chemistry Experiment – Fourier Transform Spectrometer (ACE-FTS) and surface flask measurements from 14 National Oceanic and Atmospheric Administration – Earth System Research Laboratory (NOAA-ESRL) sites worldwide.

We find that calculating vegetative uptake using the LRU underestimates the surface seasonal cycle amplitude (SCA) in the Northern Hemisphere (NH) mid-latitudes and high latitudes by approximately 37 ppt (35 %). The inclusion of a large tropical source is able to balance the global budget, but further improvement to the SCA and phasing would likely require a flux inversion scheme.

Compared to co-located ACE-FTS OCS profiles between 5 and 30 km, TOMCAT remains within 25 ppt (approximately 5 % of mean tropospheric concentration) of the measurements throughout the majority of this region and lies within the standard deviation of these measurements. This provides confidence in the representation of atmospheric loss and surface fluxes of OCS in the model. Atmospheric sinks account for 154 Gg S of the annual budget, which is 10 %–50 % larger than previous studies. Comparing the surface monthly anomalies from the NOAA-ESRL flask data to the model simulations shows a root-mean-square error range of 3.3–25.8 ppt. We estimate the total biosphere uptake to be 951 Gg S, which is in the range of recent inversion studies (893–1053 Gg S), but our terrestrial vegetation flux accounts for 629 Gg S of the annual budget, which is lower than other recent studies (657–756 Gg S). However, to close the budget, we compensate for this with a large annual oceanic emission term of 689 Gg S focused over the tropics, which is much larger than bottom-up estimates (285 Gg S). Hence, we agree with recent findings that missing OCS sources likely originate from the tropical region.

This work shows that satellite OCS profiles offer a good constraint on atmospheric sinks of OCS through the troposphere and stratosphere and are therefore useful for helping to improve surface budget terms. This work also shows that the LRU approach is an adequate representation of the OCS vegetative uptake, but this method could be improved by various means, such as using a higher-resolution GPP product or plant-functional-type-dependent LRU. Future work will utilise TOMCAT in a formal inversion scheme to better quantify the OCS budget.

## 1   Introduction

Carbonyl sulfide (OCS) is the most abundant of all sulfur-containing gases in the atmosphere and is important due to its potential use as a proxy for the photosynthetic uptake of carbon dioxide ($CO_2$) by vegetation (Sandoval-Soto et al., 2005; Montzka et al., 2007; Campbell et al., 2008; Suntharalingam et al., 2008; Blonquist et al., 2011; Berry et al., 2013; Launois et al., 2015b). Furthermore, due to its oxidation in the stratosphere, OCS is the largest source of sulfuric acid in the stratospheric aerosol layer in times of low volcanic activity (Crutzen, 1976; Kremser et al., 2016). In the troposphere, OCS has a global mean mixing ratio (mole fraction) of approximately 480 parts per trillion (ppt) and a lifetime of approximately 2.5 years (Montzka et al., 2007). In the stratosphere the OCS mixing ratio declines strongly with increasing altitude due to photochemical removal and the only source being transport from the troposphere. The stratospheric lifetime is approximately $64 \pm 21$ years (Barkley et al., 2008), ranging from $54.1 \pm 9.7$ years in the sub-tropics to $103.4 \pm 18.3$ years in the Antarctic (Hannigan et al., 2022). The long stratospheric partial lifetime (defined as total atmospheric OCS burden divided by loss in stratosphere) reflects the low total mass of OCS there, giving a smaller absolute loss, and the low OH concentration compared to the troposphere.

Observations of OCS by the Network for the Detection of Atmospheric Composition Change (NDACC) using ground-based solar-viewing Fourier transform infrared spectrometers (FTIR) show weak positive trends between 2009 and 2016 in the troposphere at most of the 22 measurement sites of $< 1\,\% \, \mathrm{yr}^{-1}$ (Hannigan et al., 2022). Stronger positive trends, up to $1.93 \pm 0.26\,\% \, \mathrm{yr}^{-1}$ for 2009–2016, are observed in the stratosphere above all sites, except for the tropical sites Mauna Loa and Altzomoni, which show a negative trend (approximately $-0.5\,\% \, \mathrm{yr}^{-1}$). Furthermore, a downturn in free-tropospheric OCS concentration reveals a negative trend between 2016 and 2020 at all sites (Hannigan et al., 2022). Kremser et al. (2015) showed positive OCS trends between 2001 and 2015 determined from ground-based FTIR total column measurements at three Southern Hemisphere (SH) sites (also used by Hannigan et al., 2022) and driven by changes in the tropospheric column. In contrast, the National Oceanic and Atmospheric Administration – Earth System Research Laboratory (NOAA-ESRL)

global monitoring network has 14 sites and shows no consistent trend in surface OCS at any one location during the period of 2000 to 2005 (Montzka et al., 2007). Additionally, Glatthor et al. (2017) concluded that the tropospheric OCS budget is balanced based on a global Michelson interferometer for passive atmospheric sounding (MIPAS) satellite dataset (2002–2012), while ground-based partial-column measurements at the Jungfraujoch ($46.5°\,$N, $8.0°\,$E) showed no significant trend in the free troposphere between 2008 and 2015 (Lejeune et al., 2017).

One of the main sources of atmospheric OCS is oceanic emission, with total estimates ranging from 230 to $992\,\mathrm{Gg}\,\mathrm{S}\,\mathrm{yr}^{-1}$ (Kettle et al., 2002; Montzka et al., 2007; Suntharalingam et al., 2008; Berry et al., 2013; Glatthor et al., 2015; Kuai et al., 2015; Launois et al., 2015a; Lennartz et al., 2021; Ma et al., 2021; Remaud et al., 2022). Oceanic emission has three main sources: direct emission of OCS, oxidation of emitted dimethyl sulfide (DMS) and oxidation of emitted carbon disulfide ($CS_2$). Both light-dependent (photochemical) and light-independent production play a role in oceanic emission (Launois et al., 2015a), the former linked primarily to incident UV radiation at the sea surface and the latter far below the surface. Both are driven by biological production and are proportional to amounts of chromophoric dissolved organic matter (CDOM), especially at the surface where it can act to absorb some of the available light (Lennartz et al., 2021). Furthermore, Lennartz et al. (2021) suggest the importance of direct ocean-emitted OCS and oxidised $CS_2$ exceeds that of oxidised DMS, which accounts for only a small portion of the overall ocean-borne OCS emissions.

Vegetative uptake is the most important sink of atmospheric OCS, and its magnitude is significantly more uncertain than the ocean flux, with estimates ranging from 210 to $2400\,\mathrm{Gg}\,\mathrm{S}\,\mathrm{yr}^{-1}$ (Kettle et al., 2002; Sandoval-Soto et al., 2005; Suntharalingam et al., 2008; Berry et al., 2013; Glatthor et al., 2015; Kuai et al., 2015; Launois et al., 2015b; Kooijmans et al., 2021; Ma et al., 2021; Maignan et al., 2021; Remaud et al., 2022). OCS is consumed during the photosynthesis process, which proceeds along the same enzymatic pathways as $CO_2$ (Protoschill-Krebs et al., 1996). However, unlike for $CO_2$, this process is one-way due to the irreversible OCS hydrolysis reaction, catalysed by carbonic anhydrase (CA) (Protoschill-Krebs et al., 1996). OCS hydrolysis also occurs in soil, primarily catalysed by carbonic

anhydrase contained in bacteria and fungi (Kesselmeier et al., 1999; Smith et al., 1999; Li et al., 2005; Seibt et al., 2006; Kato et al., 2008), as well as by other enzymes, such as nitrogenase, CO dehydrogenase and $CS_2$ hydrolase (Smith and Ferry, 2000; Masaki et al., 2021). Soil uptake is the second-largest OCS sink, with an estimated annual net loss of 30–355 Gg S (Kettle et al., 2002; Montzka et al., 2007; Berry et al., 2013; Glatthor et al., 2015; Kuai et al., 2015; Kooijmans et al., 2021; Abadie et al., 2022). Other findings suggest that the seasonal variation in OCS soil uptake is relatively weak in boreal forest regions but shows dependency on soil moisture (Sun et al., 2018). Soil has also been observed to act as an emitter of OCS in certain conditions, dependent on such components as temperature, soil moisture, nitrogen content and incident solar radiation (Whelan et al., 2013; Maseyk et al., 2014; Spielmann et al., 2019; Kitz et al., 2020).

Chin and Davis (1993) presented one of the first attempts at quantifying the global OCS (and $CS_2$) budget terms, but these were subject to substantial uncertainties. However, multiple terms, such as atmospheric loss and volcanism, were subsequently used in the estimates presented by Watts (2000) and Kettle et al. (2002), the latter of which has been used as a benchmark for more recent studies (Montzka et al., 2007; Suntharalingam et al., 2008; Berry et al., 2013; Glatthor et al., 2015; Kuai et al., 2015). Analysis of flask and aircraft data spanning both hemispheres by Montzka et al. (2007) have offered the most significant updates since the aforementioned studies and suggest a vegetative sink (1115 Gg S yr$^{-1}$) up to 5 times larger than the estimate (240 Gg S yr$^{-1}$) presented by Kettle et al. (2002). Due to the negligible or weakly positive atmospheric OCS trend up to 2016, this would suggest a larger source is required for balance. The general consensus is that this must originate in the tropical oceans due to measurement peaks from satellite and aircraft observations (Glatthor et al., 2015; Kuai et al., 2015) and modelling estimates pointing to this region as an underestimated source (Berry et al., 2013; Launois et al., 2015a; Remaud et al., 2022). There is opposition from Lennartz et al. (2017, 2021), who estimate global oceanic emissions to be approximately 285–345 Gg S yr$^{-1}$, derived using a global oceanic box model and measurements of surface waters, which is a value that is too low to account for this difference entirely. A recent study has also suggested there is an underestimation in previous gridded anthropogenic OCS flux inventories of 200 Gg S yr$^{-1}$, which could account for some of the deficit (Zumkehr et al., 2018); this is supported by measurements of OCS in firn air and ice core samples (Aydin et al., 2020). Top-down estimates by Ma et al. (2021), using an inversion scheme that assimilates surface flask observations, point to a tropical source of unknown origin, but the inversion setup presented by Remaud et al. (2022) suggests a large tropical OCS source of oceanic origin. Both studies downplay the likelihood of it being of exclusively oceanic origin; hence, there is still substantial uncertainty in several of the global surface fluxes of OCS. These recent studies quantifying OCS

flux inventories show less uncertainty than previous ones. However, to improve the inventories further, increased spatial coverage by ground-based and remote atmospheric OCS observations are required, as well as OCS flux measurements (Whelan et al., 2018).

In this study, we add a further model (TOMCAT) to those already employed to simulate global OCS distribution, with emphasis on a full vertical comparison extending through the troposphere and stratosphere (approximately 5–35 km). Three inventories of fluxes are used to drive the model in separate experiments, offering different perspectives. Firstly, we created a control setup using fluxes from Kettle et al. (2002), the results of which are denoted TOMCAT$_{CON}$. Secondly, we constructed an inventory using modified fluxes from Kettle et al. (2002) in addition to a vegetative uptake flux quantified using gross primary productivity (GPP) in the leaf relative uptake (LRU) approach (Campbell et al., 2008; Stimler et al., 2012; Asaf et al., 2013) and one flux from the literature, i.e. anthropogenic emissions from Zumkehr et al. (2018). The model simulation using this array of fluxes is referred to as TOMCAT$_{OCS}$. Finally, we compiled an inventory using newly available bottom-up fluxes from recent literature, TOMCAT$_{SOTA}$ (TOMCAT$_{state-of-the-art}$). Each inventory of OCS fluxes is used in the TOMCAT 3-D chemical transport model (CTM) over the time period 2004–2018, providing fresh insight into the magnitude and location of the fluxes of OCS and how this translates vertical information of OCS into improved understanding of both surface and atmospheric fluxes. Furthermore, to investigate differences in SH stratospheric mixing ratios between TOMCAT$_{OCS}$ and satellite observations, additional simulations were performed to assess the influence of a hypothetical reduction in stratospheric photolysis would have on OCS distribution.

Section 2 summarises the data used for evaluating the model. The model setup and each flux inventory are described in Sect. 3. Results and comparisons with tropospheric and stratospheric satellite observations from the Atmospheric Chemistry Experiment infrared Fourier transform spectrometer instrument (Bernath, 2017; Boone et al., 2020) and measurements made by the NOAA-ESRL flask network (Montzka et al., 2007) are shown in Sect. 4 and discussed further in Sect. 5. Concluding remarks are presented in Sect. 6.

## 2 Observations

### 2.1 Atmospheric Chemistry Experiment – Fourier Transform Spectrometer Observations

Onboard SCISAT (Science Satellite), launched in August 2003, is the Atmospheric Chemistry Experiment infrared Fourier transform spectrometer (ACE-FTS), which operates in a solar occultation mode measuring radiation between 750 and 4400 cm$^{-1}$ at a spectral resolution of 0.02 cm$^{-1}$ (Bernath et al., 2005; Bernath, 2017). Although the planned SCISAT mission duration was only 2 years, it now has a data record

spanning 19 years. This longevity makes the ACE-FTS a valuable tool for measuring atmospheric trace gases and characterising their variability and trends. Atmospheric trace gas profiles are retrieved using a non-linear least-squares global-fit approach on the measurement altitude grid (3 km vertical resolution) and then interpolated onto a uniform 1 km grid. ACE-FTS is capable of measuring profiles for a number of trace gases, including OCS, from 5 km (or cloud top) up to about 30 km. OCS is retrieved using microwindows of various widths between 2039.01 and 2057.52 cm$^{-1}$, including a band at 1950.10 cm$^{-1}$ to minimise the impact of $H_2O$ interference. Because the primary science mission of ACE-FTS is to measure atmospheric ozone distributions over Canada, the satellite's orbit is such that approximately 60 % of all measurements are at latitudes poleward of $\pm 60°$. However, over the course of a year measurements are taken over a wide range of latitudes, providing a wealth of data with which to validate global CTM simulations. For this study, ACE-FTS version 4.1 (hereafter ACE) retrieved profiles from February 2004 to December 2018 (approximately 98 000 profiles) (Boone et al., 2020) that are used here in the validation of the modelled TOMCAT OCS distribution. The version 4.1 retrievals incorporate a new instrumental line shape (Boone and Bernath, 2019) and utilise the 2016 HIgh-Resolution TRANsmission molecular absorption database (HITRAN) data (Gordon et al., 2017). Systematic errors in OCS measurements occur as a result of contamination from other gases in the microwindow (and clouds), while random errors are induced by random fitting errors from the least-squares analysis. Both of these have generally been improved in the version 4.1 product over version 3.6 (Boone et al., 2020). The errors in ACE OCS measurements amount to a mean of approximately 3.8 % throughout the entire profile globally. In the lower troposphere, below 10 km, errors are larger at approximately 7.2 %, and above 20 km they are relatively low at 3.4 %.

## 2.2 NOAA-ESRL flask measurements

The surface OCS measurements described here are shown in Sect. 4; here we present a summary of the method of data collection (performed by the NOAA-ESRL network) and the site information (see Table 1). Flasks of ambient air have been collected approximately 1 to 5 times per month at 14 measurement sites across both hemispheres since early 2000. Measurements of the OCS concentrations within the flasks are made using gas chromatography and mass spectrometry at the NOAA-ESRL Boulder laboratories (Montzka et al., 2007). In this study, we use data from all the Halocarbons and other Atmospheric Trace Species (HATS) surface measurement sites for the purpose of validating the surface OCS concentrations from the TOMCAT model.

## 3 Chemical transport modelling of OCS

### 3.1 TOMCAT model setup

We have used the TOMCAT 3-D off-line CTM (Chipperfield, 2006; Monks et al., 2017) to model atmospheric OCS. This model has been used in a wide range of studies, including to better constrain methane flux estimations (Wilson et al., 2016; Parker et al., 2018), to provide a forward model for methane flux inversions (McNorton et al., 2018) and to investigate stratospheric ozone depletion (Claxton et al., 2019). In this work, TOMCAT is driven by meteorological reanalysis data (ERA-Interim) from the European Centre for Medium-Range Weather Forecasts (ECMWF; Dee et al., 2011). ERA-Interim convective mass fluxes are used following the scheme presented in Feng et al. (2011). The model distribution of OH is specified from pre-computed fields that vary monthly but not inter-annually. The monthly distributions are taken from Spivakovsky et al. (2000) and scaled by a factor of 0.92 in accordance with Huijnen et al. (2010). The photolysis loss is based on precomputed rates from the full chemistry version of TOMCAT (Monks et al., 2017). Atmospheric OH loss accounts for approximately 120–130 Gg S yr$^{-1}$ (roughly 10 % of the total OCS sink) of the TOMCAT$_{OCS}$ budget, and photolysis accounts for about a quarter of this, i.e. 30–34 Gg S yr$^{-1}$ (approximately 3 % of the total OCS sink). TOMCAT$_{OCS}$ and TOMCAT$_{SOTA}$ are spun for 10 years prior to 2004 and then run between 2004 and 2018 at a horizontal resolution of approximately $2.8° \times 2.8°$ (T42 Gaussian grid), with 60 atmospheric layers from the surface up to 0.1 hPa on a time step of 6 h. In the case of TOMCAT$_{OCS}$, the vegetative flux is also calculated every 6 h. TOMCAT$_{CON}$ is initialised using the distribution of TOMCAT$_{OCS}$ at the end of the spin-up period but is run for just a single year, 2004, due to the negative trend and its purpose in this study as a point of reference, rather than a benchmark for improvement. Surface flux fields of OCS were implemented within TOMCAT on a monthly $1° \times 1°$ grid. Depending on the inventory in use, some vary inter-annually, i.e. vegetative uptake and anthropogenic emission, and the remaining fluxes do not (oceanic emission, soil uptake and biomass burning). These are mapped onto the model grid in a way that conserves local distributions and the total global flux. For comparison with ACE, the geopotential height output from the model is converted to altitude; this is done using the hypsometric equation at a reference pressure of 1000 hPa, and it is then interpolated onto the 1 km equidistant altitude grid used by ACE-FTS. Furthermore, the profiles outputted by TOMCAT are spatio-temporally co-located with the ACE observations to provide a precise like-for-like comparison. Monthly mean surface concentrations are calculated from the flask observations made by the NOAA-ESRL network and compared with the monthly mean TOMCAT output averaged across the time period used for each respective setup. Co-sampling of the model output with NOAA-ESRL mea-

**Table 1.** NOAA-ESRL flask sampling site information for OCS measurements (Montzka et al., 2007).

| Code | Name | Country | Latitude (° N) | Longitude (° E) | Elevation (metres) |
|------|------|---------|------|------|------|
| ALT | Alert, Nunavut | Canada | 82.5 | −62.5 | 185 |
| BRW | Utqiaġvik (formerly Barrow), Alaska | United States | 71.3 | −156.6 | 11 |
| CGO | Kennaook/Cape Grim, Tasmania | Australia | −40.7 | 144.7 | 94 |
| HFM | Harvard Forest, Massachusetts | United States | 42.5 | −72.2 | 340 |
| KUM | Cape Kumukahi, Hawaii | United States | 19.6 | −155.0 | 8 |
| LEF | Park Falls, Wisconsin | United States | 45.9 | −90.3 | 472 |
| MHD | Mace Head, County Galway | Ireland | 53.3 | −9.9 | 5 |
| MLO | Mauna Loa, Hawaii | United States | 19.5 | −155.6 | 3397 |
| NWR | Niwot Ridge, Colorado | United States | 40.1 | −105.6 | 3523 |
| PSA | Palmer Station | Antarctica (United States) | −64.8 | −64.1 | 10 |
| SMO | Tutuila | American Samoa | −14.2 | −170.6 | 42 |
| SPO | South Pole | Antarctica (United States) | −90.0 | −24.8 | 2810 |
| SUM | Summit | Greenland | 72.6 | −38.4 | 3210 |
| THD | Trinidad Head, California | United States | 41.1 | −124.2 | 107 |

surements would be a more representative comparison, but here we have not subsampled the model on the specific days of NOAA observations.

## 3.2 Kettle flux inventory

The fluxes described in this section originate from the literature (Watts, 2000; Kettle et al., 2002; Suntharalingam et al., 2008) and are used to run the control simulation, TOMCAT$_{CON}$. This model run is utilised as a comparison to the model driven by our new inventory of fluxes described in Sect. 3.3, TOMCAT$_{OCS}$ and to TOMCAT$_{SOTA}$ in Sect. 3.4. TOMCAT$_{CON}$ was initialised using OCS values in each grid box from TOMCAT$_{OCS}$ after 10 years (1994–2003) spin-up and run for only a single year (2004) due to the net negative budget from these fluxes of approximately $-46$ Gg S yr$^{-1}$.

Three of the six sources used to simulate TOMCAT$_{CON}$ are oceanic: a direct OCS flux term, one due to oxidation of CS$_2$ and one due to oxidation of DMS. These were converted to OCS emissions using molar conversion factors (Chin and Davis, 1993; Barnes et al., 1994). The OCS and CS$_2$ emission terms were quantified using a physio-chemical model, with the main source being from photochemical production (Kettle et al., 2002). However, as DMS measurements are more abundant than OCS and CS$_2$, these were used to parameterise this flux (Kettle and Andreae, 2000). Anthropogenic OCS emissions consist of two factors, a direct term and one from the oxidation of CS$_2$, the latter being considerably larger. They are both calculated here using SO$_2$ fields from Watts (2000) due to the extensive datasets available and a relationship between the facilities that release SO$_2$ and OCS, despite there being no direct chemical reaction (Kettle et al., 2002). The final source term is biomass burning scaled similarly to that in Kettle et al. (2002) but varied according to the monthly climatology of Duncan et al. (2003).

The three sink terms are an oceanic sink, soil uptake and a vegetative sink. The first was quantified using the same physio-chemical model used for the OCS and CS$_2$ source terms described above and covers the periods in the year where the direct oceanic emission of OCS flips from being a source to becoming a sink. These are focused mostly over extra-tropical open-ocean regions and during each hemisphere's summer period. Gridded soil uptake was calculated by applying correction factors for temperature, ambient OCS and soil water content to a standardised uptake rate of 10 pmol m$^{-2}$ s$^{-1}$ (Kesselmeier et al., 1999). The monthly mean climatological data for the temperature and soil water content is taken from Sellers et al. (1995), where the soil water content is a percentage of saturation in the top 2 cm of soil. Anoxic soil emissions are neglected in this study, but with the availability of new datasets, future simulations could include these sources (Abadie et al., 2022; Whelan et al., 2022). Finally, the vegetative uptake is calculated by employing a normalised difference vegetation index (NDVI) to scale net primary productivity (NPP) distribution from Fung et al. (1987). We also scale up this term to the quoted upper limit of 270 Gg S yr$^{-1}$ by Kettle et al. (2002). As mentioned in Sect. 3.1, removal of atmospheric OCS by OH loss and photolysis is also accounted for in the model.

The spatial distribution for the months of January, April, July and October for the vegetation uptake, soil uptake and oceanic emissions used in TOMCAT$_{CON}$ are presented in the Supplement in Figs. S1, S2 and S3, respectively.

## 3.3 LRU approach and modified flux inventory

TOMCAT$_{OCS}$ uses an array of fluxes that is orientated around a calculated vegetative uptake term, $F_{OCS}$, using the LRU approach (Campbell et al., 2008; Stimler et al., 2012; Asaf et al., 2013). The calculation of $F_{OCS}$ is explained in

Sect. 3.3.1, and a summary of the accompanying fluxes is provided in Sect. 3.3.2, including the full budget in Table 2.

### 3.3.1 Calculating OCS vegetative uptake using gross primary productivity

The new vegetative sink calculation used in TOMCAT$_{OCS}$ differs fundamentally from the method described in Sect. 3.2 and used in the control model simulation as the use of NPP has been shown to underestimate the seasonal amplitude in other modelling studies (Suntharalingam et al., 2008; Berry et al., 2013). Sandoval-Soto et al. (2005) suggested that using NPP to calculate OCS uptake would underestimate the global burden, and therefore they recommend using gross primary productivity (GPP) as an alternative. Furthermore, they were the first to quantify deposition velocity ratios for $CO_2$ and OCS for different plant types, which previous studies had assumed to be equal.

$$F_{OCS} = GPP \frac{[OCS]}{[CO_2]} \times LRU \qquad (1)$$

Using Eq. (1) we calculated the vegetative flux of OCS ($F_{OCS}$), in units of Gg S yr$^{-1}$, by scaling GPP (Gg C yr$^{-1}$) using a LRU of 1.6, which is a mean value from gas exchange measurements of 22 plant species (Stimler et al., 2012). LRU is the ratio of OCS assimilation rates to $CO_2$ at the leaf scale, both normalised by their respective mixing ratios, signified by the square brackets in Eq. (1), in units of parts per billion. Fluxes, including $F_{OCS}$, were implemented in TOMCAT (in units of molec. cm$^{-2}$ s$^{-1}$). At each time step in the model, a new $F_{OCS}$ value is calculated, as [OCS] is the mixing ratio from the previous time step, starting at a value of 500 ppt in 1994. The use of a constant LRU value was found to contribute less to errors in the calculation of $F_{OCS}$ than differences in GPP between models on a continental scale by Hilton et al. (2017). However, Maignan et al. (2021) found the opposite, 70 % of uncertainty was attributed to the use of three different LRU datasets, while consideration of different land surface models introduced an uncertainty of 40 %. As there are available plant-functional-type-dependent LRU datasets, implementing spatially varying LRU values will be undertaken in future work (Seibt et al., 2010; Maignan et al., 2021). The GPP flux used in our calculation, generated by the Joint UK Land Environment Simulator (JULES) model, applies the WATer and Global CHange (WATCH) Forcing Data methodology to ERA-Interim reanalysis (WFDEI) between 1979 and 2012 and uses Global Precipitation Climatology Centre (GPCC) precipitation data (Slevin et al., 2016). Monthly mean gridded $CO_2$ surface mixing ratios, used in the calculation of $F_{OCS}$, came from a TOMCAT simulation that assimilated surface flask concentrations for 2010 (see Fig. S4 in the Supplement) (Gloor et al., 2018). Given that 2010 is situated approximately in the middle of the study period, it should be a reasonable estimate of the long-term average. Therefore, the GPP data used was also for 2010

given its relatively small inter-annual variability (Chen et al., 2017). As we compare only monthly means at the surface and seasonal OCS to ACE, long-term inter-annual variability was not considered in the scope of this work. However, future work using TOMCAT can exploit longer-term records of surface $CO_2$ mixing ratios and GPP. Our resulting estimate of the mean global yearly value of $F_{OCS}$ between 2004 and 2018 is 629 Gg S, which is nearly 3 times the value of Kettle et al. (2002) at 240 Gg S, over half that of the largest estimation of 1115 Gg S by Montzka et al. (2007) in Table 2, and under half that estimated by Launois et al. (2015b) of 1335 Gg S yr$^{-1}$ using the Organising Carbon and Hydrology In Dynamic Ecosystems (ORCHIDEE) land surface model. The spatial distribution of $F_{OCS}$ for the months of January, April, July, and October in 2010 only is presented in the Supplement in Fig. S5.

### 3.3.2 Scaling of OCS prior fluxes to balance OCS budget

As the fluxes described in Sect. 3.2 are utilised in constructing the inventory for TOMCAT$_{OCS}$, with the exception of calculating the vegetative uptake and anthropogenic emissions (which are taken from Zumkehr et al., 2018), the fluxes must be modified to suitably close the overall budget, which we assume to be in balance due to a negligible or weak trend in the majority of the study period (Montzka et al., 2007; Kremser et al., 2015; Glatthor et al., 2017; Lejeune et al., 2017; Hannigan et al., 2022). As $F_{OCS}$ is larger than the vegetative uptake term than that of Kettle et al. (2002), we scale up several of the emission terms described in Sect. 3.2; however, some of the difference was accounted for in the larger anthropogenic emissions. Furthermore, some of the fluxes were adjusted to better represent recent estimations in the literature, i.e. soil uptake.

TOMCAT$_{OCS}$ makes use of anthropogenic OCS emissions presented by Zumkehr et al. (2018). Like Kettle et al. (2002), anthropogenic OCS emissions consist of two factors, a direct term and one from the oxidation of $CS_2$, the latter being considerably larger. A total of 11 anthropogenic sources of OCS were quantified by Zumkehr et al. (2018) for the period 1980–2012, with the largest contributions originating from residential and industrial coal usage and the rayon industry. Emission factors for each source are applied to country-scale industrial activity data obtained from a wide range of sources and then gridded spatially and temporally based on a gridded proxy flux (Zumkehr et al., 2018).

Scaling OCS emitted from biomass burning (as described in Sect. 3.2) and anthropogenic sources was not considered suitable to balance increases in sink terms, as these are less uncertain than oceanic emissions. Furthermore, biomass burning is more focused in lower-latitude agricultural regions, and anthropogenic emissions tend to be focused over point sources, mostly in Asia.

**Table 2.** Global OCS budgets (units Gg S yr$^{-1}$). Values for past studies are an average of the upper and lower limits stated in those studies unless a value is stated exactly. Values for this study are an average between 2004 and 2018.

| Source or sink process | Kettle et al. (2002) | Montzka et al. (2007) | Suntharalingam et al. (2008) | Berry et al. (2013)[a] | Ma et al. (2021)[b] | Remaud et al. (2022) | TOMCAT$_{OCS}$ fluxes |
|---|---|---|---|---|---|---|---|
| Vegetation | −238 | −1115 | −490 | −738 | −1053 | −657 | −629 |
| Oxic soil | −130 | −127 | −120 | −355 | | −236 | −322 |
| Reaction with OH | −94 | −96 | | −101 | −101 | −100 | −122 |
| Reaction with O($^1$D) | −11 | −11 | −130 | 0 | 0 | 0 | 0 |
| Photolysis | −16 | −16 | | 0 | −40 | 0 | −32 |
| Ocean | 0 | 0 | 0 | 0 | 0 | 0 | −39 |
| Total sinks | −489 | −1365 | −740 | −1194 | −1194 | −993 | −1144 |
| Ocean (OCS) | 41 | 40 | | | 40 | | |
| Ocean (CS$_2$) | 84 | 240 | 230 | 876 | 81 | 269 | 689 |
| Ocean (DMS) | 154 | | | | 156 | | |
| Total ocean emission | 279 | 280 | 230 | 876 | 277 | 269 | 689 |
| Anthropogenic (OCS) | 64 | 64 | | 64 | 155 | | |
| Anthropogenic (CS$_2$) | 116 | TS1 | 180 | 116 | 188 | 398 | 410 |
| Anthropogenic (DMS) | 1 | 0 | | 1 | 6 | | 0 |
| Total anthropogenic emission | 181 | 64 TS2 | 180 | 181 | 349 | 398 | 410 |
| Biomass burning | 38 | 106 | 70 | 136 | 136 | 53 | 42 |
| Other (mainly wetlands and anoxic soils) | 26 | 66 | 25 | 0 | 0 | 0 | 0 |
| Total sources | 523 | 516 TS3 | 505 | 1193 | 762 | 720 | 1141 |
| Net budget | 34 | −849 TS4 | −235 | −2 | −432 | −273 | −3 |

[a] Ocean emission term includes an additional photochemical oceanic flux of 600 Gg S. [b] Posterior estimates from the Su inversion are shown here. An "unknown" term is accounted for in the net budget, which was optimised spatially and temporally using an inverse system.

The soil flux utilised for TOMCAT$_{OCS}$ was calculated by Kettle et al. (2002) using the method described in Sect. 3.2 and assumes a constant 500 ppt OCS ambient value in the scaling of the standardised uptake. Soil uptake was scaled by 2.5 times from 130 to 322 Gg S yr$^{-1}$ to bring it in line with literature findings that estimate soil uptake to be between 236–507 Gg S yr$^{-1}$ (Berry et al., 2013; Launois et al., 2015b; Ma et al., 2021; Remaud et al., 2022). These studies used different approaches. Berry et al. (2013) use a global carbon cycle model, SiB 3, to obtain a new estimate of soil uptake based on empirical data and a mechanistic understanding of the processes influencing OCS diffusion into soil. Launois et al. (2015b) use H$_2$S soil deposition to infer OCS fluxes, as this is a byproduct of the OCS hydrolysis reaction and therefore a proxy for OCS uptake. Ma et al. (2021) and Remaud et al. (2022) use inverse frameworks, the former estimate a combined vegetative and soil uptake of 1053 Gg S yr$^{-1}$,

while the latter estimate a soil uptake of 236 Gg S yr$^{-1}$. Recent work using mechanistic soil uptake models (Ogée et al., 2016) suggest oxic soil uptake is lower than the estimates discussed here. Kooijmans et al. (2021) estimate an annual uptake of 89 Gg S yr$^{-1}$, and Abadie et al. (2022) estimate 126 Gg S yr$^{-1}$. These values do not yet align with inversion studies, adding to the uncertainty in surface fluxes, especially in the tropics, that accounts for a large portion of terrestrial OCS uptake (Ma et al., 2021; Remaud et al., 2022).

Initial testing of our new fluxes in TOMCAT yielded low-biased simulated OCS concentrations at Northern Hemisphere (NH) NOAA-ESRL sites, ALT, BRW and MHD but a seasonal cycle with appropriate amplitude (not shown). To improve the agreement, the direct and indirect OCS ocean emissions arising from DMS were increased by a factor of 2. These fluxes were chosen as their spatial distribution includes peaks in the northern Atlantic and Pacific re-

gions. When including the reduction implemented for these terms in the Southern Ocean, the global net increase for direct OCS and indirect OCS from DMS is roughly 10 and $7\,\mathrm{Gg\,S\,yr^{-1}}$, respectively, which is relatively small compared to the changes to the vegetative and soil OCS fluxes.

Suntharalingam et al. (2008) recommend a reduction in direct OCS and indirect OCS emissions from DMS by 40 % (as this yielded the smallest root-mean-squared error in their analysis) in SH mid-latitude (ML) and high-latitude (HL, here defined as 60–90°) regions, due to the resulting improvements to the seasonal cycle at Antarctic NOAA-ESRL sites. They also implemented an enhanced OCS tropical ocean source that was aseasonal and uniform across the tropics. However, here we scale up the $CS_2$ source term to $439\,\mathrm{Gg\,S\,yr^{-1}}$ to balance the increased vegetation and soil sink terms discussed above and bring the net budget to near balance. We scaled this flux not necessarily because it was suspected that $CS_2$ was the erroneous term in the OCS budget but because it is more realistic to add a flux that is focused spatially over the tropical region already (Kuai et al., 2015). The reason for this geographical distribution is that $CS_2$ emissions are proportional to temperature and incident solar radiation, and this is why the tropics show the strongest emissions (Kettle et al., 2002). Bottom-up estimates of global annual direct and indirect oceanic emissions total approximately 285–345 $\mathrm{Gg\,S\,yr^{-1}}$ (Lennartz et al., 2017, 2021). This is thus not enough to account for the discrepancy in the global OCS budget. However, for the purposes of this study, we allocate the discrepancy into oceanic emissions due to the co-location of $CS_2$ emission fields over the tropics, and this is the most suitable representation.

Using the flux inventory described here, TOMCAT$_{OCS}$ simulations were carried out covering 2004 to 2018, initialised at 500 ppt in every grid box, and spun up for 10 years between 1994 and 2004. Average yearly burdens for 2004 to 2018 yield a broadly closed OCS budget. Inter-annual variability in meteorology will have an impact on the model's ability to have a mean closed budget over the full time period. The vegetative flux sits roughly in the middle of literature estimates, but the total sink term (1144 $\mathrm{Gg\,S\,yr^{-1}}$) is similar to larger estimates from Berry et al. (2013) and Ma et al. (2021), as seen in Table 2, as well as estimates from Glatthor et al. (2015), Kuai et al. (2015) and Launois et al. (2015b) that are not shown in Table 2. Atmospheric destruction, mainly in the form of tropospheric loss from OH and stratospheric photolysis reactions, account for approximately 154 $\mathrm{Gg\,S\,yr^{-1}}$ removal, which is 25 % larger than fields used in earlier studies in Table 2 of roughly 126 $\mathrm{Gg\,S\,yr^{-1}}$ derived by Watts (2000). The total oceanic emission has been increased by 146 % from the starting point of Kettle et al. (2002); the majority of this increase is focused in the tropical region. With a global net annual emission of 1141 Gg S, roughly equal to that of our sink terms, the model yields 14 years of broadly balanced OCS budget, with all terms broadly in line with the findings of recent studies (Berry et al., 2013; Glatthor et al., 2015; Kuai et al., 2015; Launois et al., 2015b; Ma et al., 2021; Remaud et al., 2022). The spatial distributions for the months of January, April, July and October for the adjusted soil uptake and oceanic emissions used in TOMCAT$_{OCS}$ are presented in the Supplement in Figs. S6 and S7, respectively.

## 3.4 State-of-the-art flux inventory

Emissions used in the TOMCAT$_{SOTA}$ simulation use five unique fluxes, which vary monthly, and unlike those used in TOMCAT$_{OCS}$ and TOMCAT$_{CON}$ they also vary inter-annually. All implemented fluxes are sourced from various bottom-up OCS inventory studies. The five sectors in use here are vegetation uptake, soil uptake, oceanic emissions, anthropogenic emissions and biomass burning emissions. Due to the biomass burning and anthropogenic emissions only being available between 2010 and 2015, 2015 fluxes are repeated through 2016–2018. The same is done for 2010 fluxes for the period of 2004 to 2009 for all five emission or sink fields.

The sink due to vegetation was derived by implementing the OCS vegetative uptake model from Berry et al. (2013) into the land surface model ORCHIDEE, undertaken and explained in detail by Maignan et al. (2021). Berry et al. (2013) calculate OCS uptake using a series of mechanistically and empirically derived conductances that quantify diffusion of OCS from the boundary layer to leaf stomata, where it is eventually hydrolysed by CA in the leaf cell. Additionally, Maignan et al. (2021) compare the mechanistic model to the LRU-GPP approach, used in the calculation of OCS in Sect. 3.3.1, by running the two in the LMDz6 atmospheric transport model. They found that while the mechanistic approach works better on shorter timescales and smaller spatial scales, both are suitable for global estimation of vegetative OCS uptake. Preliminary work on implementing a mechanistic soil uptake model, originating from Ogée et al. (2016), into ORCHIDEE was used as the soil flux in this work (Abadie et al., 2022). Calculation of both vegetation and soil uptake using ORCHIDEE utilise temporally and spatially varying OCS surface mixing ratios (Remaud et al., 2023) obtained from the TM5 atmospheric transport model and driven by posterior fluxes calculated by Ma et al. (2021). Estimates for each flux are −532 and −264 $\mathrm{Gg\,S\,yr^{-1}}$ for vegetation and soil, respectively.

Oceanic emissions constitute two parts, direct OCS and indirect $CS_2$ emissions. Direct OCS is estimated using a global box model and supplemented by measurements of OCS, where the former is developed by von Hobe et al. (2003) and further improves the quantification of the photoproduction rate and parameterisation of light-independent production and employs satellite observations of CDOM for use in the model (Lennartz et al., 2017, 2021). Indirect emissions are estimated using $CS_2$ concentration measurements at the surface and converted using a molar conversion ra-

tio of 0.81 (Chin and Davis, 1993; Lennartz et al., 2017). Biomass burning emissions are estimated by Stinecipher et al. (2019) using the Global Fire Emissions Database, version 4 (GFED4), and scaling CO emissions to OCS. GFED4 utilises six biomass burning categories: savanna and grassland, boreal forests, temperate forests, tropical deforestation and degradation, peatland fires, and agricultural waste burning. Their estimates for the period 1997–2016 total $60 \pm 37\,\mathrm{Gg\,S\,yr^{-1}}$. Finally, anthropogenic emissions are from the study by Zumkehr et al. (2018), as described in Sect. 3.3.2, which account for roughly $402\,\mathrm{Gg\,S\,yr^{-1}}$ of OCS emissions per year.

The spatial distribution for the months of January, April, July, and October (2010 only) for the vegetation uptake, soil uptake and oceanic emissions used in $\mathrm{TOMCAT_{SOTA}}$ are presented in the Supplement in Figs. S8, S9 and S10, respectively.

## 4 Results

$\mathrm{TOMCAT_{OCS}}$ and $\mathrm{TOMCAT_{CON}}$ are compared with NOAA-ESRL surface flask monthly mean measurements and monthly anomalies (monthly mean minus annual mean). $\mathrm{TOMCAT_{SOTA}}$ is compared only to monthly anomalies due to a negative trend in the budget. All three models are co-located to the nearest grid box and altitude to the measurements. $\mathrm{TOMCAT_{OCS}}$ is also co-located and compared to ACE, which has approximately 98 000 profiles in the modelled time period, all of which are filtered for outliers before analysis. ACE primarily measures the upper troposphere and stratosphere. As this region is less sensitive to surface processes, we only compare $\mathrm{TOMCAT_{OCS}}$. Furthermore, as $\mathrm{TOMCAT_{CON}}$ and $\mathrm{TOMCAT_{SOTA}}$ have a negative trend, this makes correcting for bias in both the troposphere and stratosphere and making a comparison throughout the entire profile challenging. An additional $\mathrm{TOMCAT_{OCS}}$ simulation with adjusted atmospheric photolysis for the year 2010 is presented and used to test the suitability of this change to correct a negative model bias in the SH stratosphere.

### 4.1 Seasonality of modelled OCS compared to surface flask measurements

TOMCAT simulates OCS distributions down to the surface, where the majority of OCS fluxes occur; it is therefore important that the model performs well at this level. Figure 1 compares the NOAA-ESRL surface flask measurements (black) with $\mathrm{TOMCAT_{OCS}}$ (solid blue) and $\mathrm{TOMCAT_{CON}}$ (orange) simulations. As $\mathrm{TOMCAT_{CON}}$ was only run for 2004 (see Sect. 3.1), a dashed blue line representing the year 2004 for $\mathrm{TOMCAT_{OCS}}$ is also shown ($\mathrm{TOMCAT_{OCS}}$ – 2004). Monthly standard deviation is calculated for each site and visualised using error bars associated with the observations and $\mathrm{TOMCAT_{OCS}}$. The modelled vertical layer of $\mathrm{TOMCAT_{OCS}}$ and $\mathrm{TOMCAT_{CON}}$ closest to the altitude of the measurement

site was used for closer comparison because the bottommost model layer does not necessarily correspond with the surface due to the relative coarseness of model grid boxes affecting the simulated surface topography. Figure 2 presents the monthly anomalies for all model runs, including $\mathrm{TOMCAT_{SOTA}}$, the root-mean-square error (RMSE) just for the seasonality (i.e. excluding influence of average concentration), and seasonal cycle amplitude (SCA) values for all model runs and the surface observations. Therefore, we can dissect the influence of changes to seasonality influencing RMSE to some extent. For example, SMO shows a reduction in RMSE from $\mathrm{TOMCAT_{CON}}$ (27.6 ppt) to $\mathrm{TOMCAT_{OCS}}$ (19.3 ppt) of 30 % in Fig. 1 but also a reduction in RMSE of 28 % in Fig. 2. Thus, this indicates that $\mathrm{TOMCAT_{OCS}}$ has improved the representation of not only the average concentration but also the seasonality.

Comparisons between $\mathrm{TOMCAT_{OCS}}$ and $\mathrm{TOMCAT_{CON}}$ are shown in Figs. 1 and 2 to emphasise the improvements made by the flux inventory developed in this study, and $\mathrm{TOMCAT_{SOTA}}$ is shown to present the latest bottom-up estimates of OCS fluxes. Generally, there is an improvement in RMSE across all the sites in Fig. 1, but in some cases (MHD and CGO) there is a degradation. In the case of CGO, this is attributed to an underestimation in average concentration due to RMSE improving in Fig. 2; however, at MHD $\mathrm{TOMCAT_{CON}}$ performs better in both RMSE and SCA metrics in both Figs. 1 and 2.

The fluxes used to model $\mathrm{TOMCAT_{OCS}}$ reduce the RMSE from an annual mean of 24.3 ppt in $\mathrm{TOMCAT_{CON}}$ at all sites to 21.2 ppt (an error reduction of 12.5 %) in Fig. 1. This improves to 5.1 ppt (20.4 %) if we exclude MHD, a particularly poorly represented site according to this metric. The surface observations show that OCS concentrations peak in April or May in the NH (ranging from 505 to 540 ppt) and reach a minimum in September or October (ranging from 386 to 488 ppt), which is consistent at all 10 NH sites and resembles the seasonality of $\mathrm{CO_2}$. Despite several of the NH sites being particularly far north (70–90° N), photosynthesis is still the dominant driving flux, emphasising the strength of the OCS vegetative uptake signal. Phasing of the seasonal cycle in the SH shifts several months earlier, with a peak in February and trough in August, driven by the seasonality of oceanic emissions.

SCA values presented in Fig. 2 show an improvement from a mean absolute difference from the observations of $\pm 30.5$ ppt (39.9 %) in $\mathrm{TOMCAT_{CON}}$ and $\pm 26.5$ ppt (34.8 %) in $\mathrm{TOMCAT_{OCS}}$. These metrics suggest that the flux inventory used in $\mathrm{TOMCAT_{OCS}}$ offers an improvement in capturing seasonality and observation representation at the surface. The mean absolute difference in SCA of $\mathrm{TOMCAT_{SOTA}}$ compared to NOAA-ESRL is $\pm 43.7$ ppt (57.2 %).

Relative to the flask measurements, the SCA at the eight NH continental measurement sites (top eight plots in Fig. 2) is captured better by $\mathrm{TOMCAT_{OCS}}$ than $\mathrm{TOMCAT_{CON}}$. At all sites there is some improvement in $\mathrm{TOMCAT_{OCS}}$ SCA,

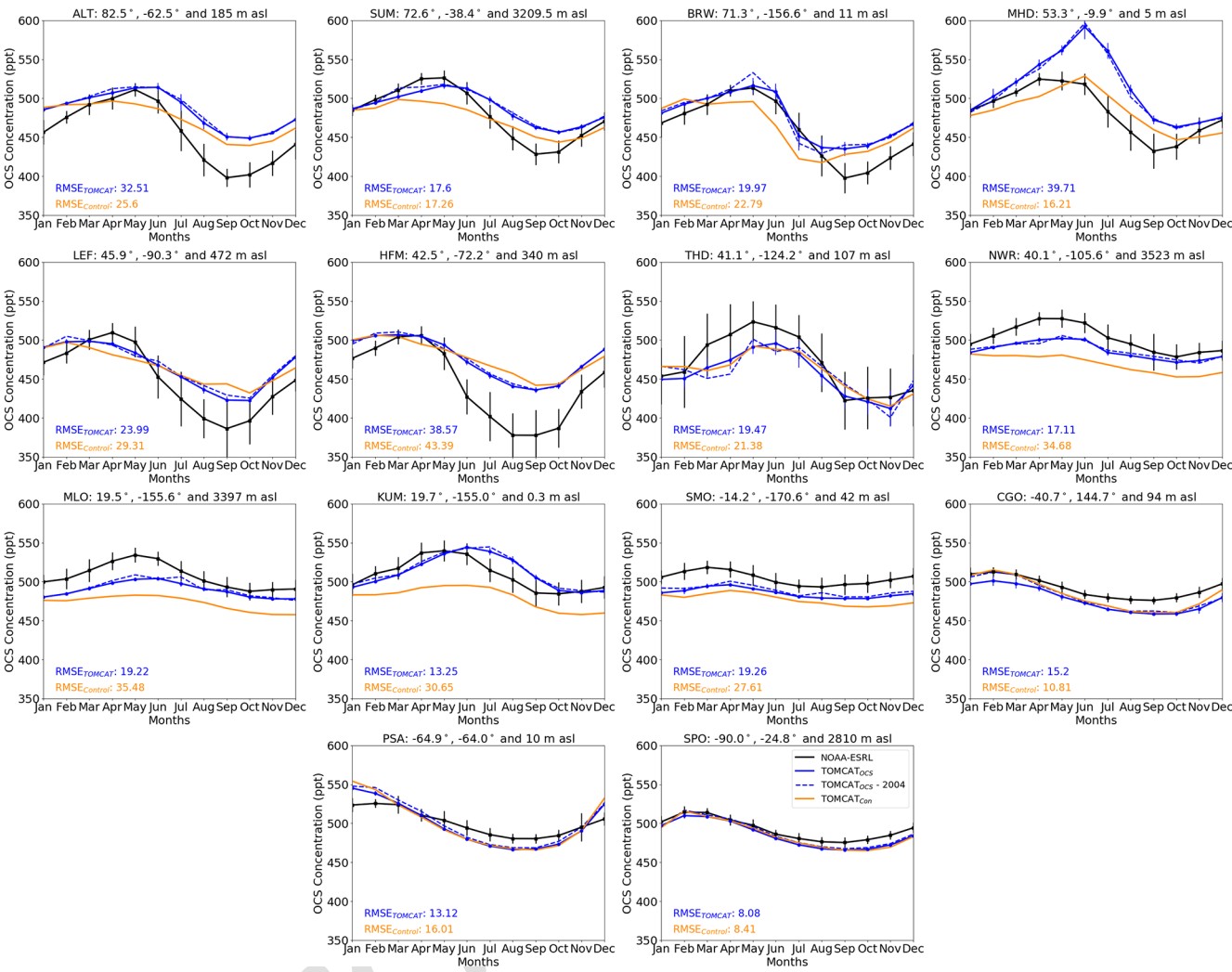

**Figure 1.** Monthly mean OCS concentration (in ppt) at NOAA-ESRL flask sites (black lines) compared with TOMCAT$_{OCS}$ (solid blue line) for the 2004 to 2018 period. The dashed blue line is just 2004 for the TOMCAT$_{OCS}$ dataset and is compared to TOMCAT$_{CON}$ (orange line). The geographical location of each site is referenced in the titles of each panel. Altitude above sea level (a.s.l.) of the site is stated, and the nearest level in TOMCAT is used for comparison. Error bars for both NOAA-ESRL and TOMCAT$_{OCS}$ represent standard deviation (units of RMSE are in ppt).

except for BRW, which shows little change but an improved RMSE (Fig. 2), and MHD which shows an overestimated SCA by 39.0 % in TOMCAT$_{OCS}$. SCA is underestimated in TOMCAT$_{CON}$ output at all 8 of these sites by approximately 38.8 ppt on average. TOMCAT$_{OCS}$ improved this disparity to an absolute difference of 36.9 ppt (MHD is overestimated). TOMCAT$_{SOTA}$ shows an absolute difference in SCA to NOAA-ESRL of 45.4 ppt, which improves substantially to 26.9 ppt if LEF and HFM are ignored. Neglecting these same sites for TOMCAT$_{OCS}$ and TOMCAT$_{CON}$, we see that TOMCAT$_{SOTA}$ shows the best performance in SCA compared to NOAA-ESRL at the continental NH sites.

LEF and HFM are dense woodland sites and have particularly large SCA that can often be a challenge for models to simulate. Here we show SCAs from TOMCAT$_{OCS}$ of

76 ppt at LEF and 71 ppt at HFM, compared to observed values of 123 and 128 ppt, respectively. The underestimation in TOMCAT$_{OCS}$ could potentially be attributed to using a single value for the LRU parameter globally, as this value is known to vary significantly between plant types (Stimler et al., 2012). The method of estimating OCS uptake using LRU clearly underestimates OCS uptake over dense vegetation, as the model is likely too coarse to include the heavily depleted OCS concentration near the surface. As the GPP product has been compared to a spatially gridded GPP from the FLUXNET network, including Harvard Forest and Park Falls, it is unlikely an underestimation in GPP in this geographical region (Slevin et al., 2016). TOMCAT$_{SOTA}$ underestimates SCA at both sites by approximately 100 ppt, high-

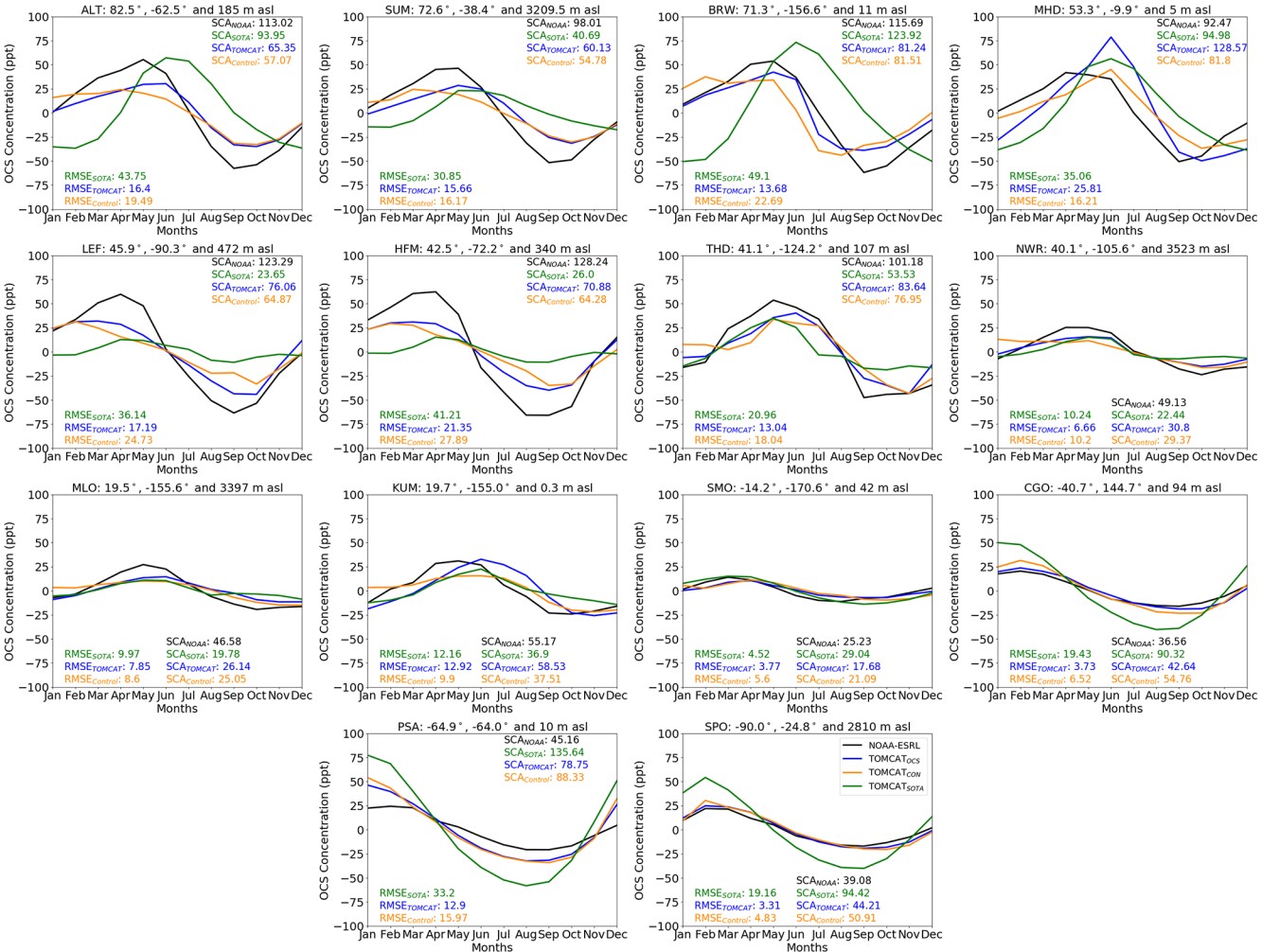

**Figure 2.** Monthly mean OCS anomalies (monthly mean less annual mean, in ppt) at NOAA-ESRL flask sites (black line) compared with TOMCAT$_{OCS}$ (blue line), TOMCAT$_{CON}$ (orange line) and TOMCAT$_{SOTA}$ (green line) for the 2004 to 2018 period. The geographical location of each site is referenced in the titles of each panel. Altitude above sea level (a.s.l.) of the site is stated, and nearest level in TOMCAT is used for comparison (units of RMSE and SCA are in ppt).

lighted by RMSE values of 36.1 and 41.2 ppt, respectively, in Fig. 2.

ALT, SUM and BRW are located at high northern latitudes, where the landscape has significantly less vegetation and is more homogeneous than at LEF and HFM, although the seasonal cycle is still driven by typical NH processes. Phasing of the peak and trough of the annual seasonal cycles at ALT, SUM and BRW is improved in TOMCAT$_{OCS}$ output compared to TOMCAT$_{CON}$, but RMSE and SCA are not improved significantly in Fig. 2. TOMCAT$_{SOTA}$ improves SCA at ALT, BRW and MHD (9.9 ppt absolute difference to NOAA-ESRL SCA) compared to TOMCAT$_{OCS}$ (39.4 absolute difference to NOAA-ESRL SCA). However, TOMCAT$_{SOTA}$ exhibits larger RMSE values in Fig. 2 due to poor phasing of the seasonality. THD and MHD are both coastal sites, and the misalignment in observed and modelled seasonal cycles is attributed to the impact from the ocean

fluxes in adjacent model grid boxes. Additionally, capturing the seasonal cycle of trace gases at a site such as MHD can be particularly challenging as there are significant seasonal changes in advected air masses. Berry et al. (2013) show overestimated peak concentration at MHD in their adjusted flux model runs, but simulations using posterior flux estimates by Ma et al. (2021) show a good alignment at this site, suggesting that the underlying cause is poorly represented surface fluxes.

At the particularly high-altitude sites of NWR and MLO, Fig. 1 shows that TOMCAT$_{OCS}$ underestimates the average concentration, additionally showing no significant improvement in SCA from TOMCAT$_{CON}$. The measured SCA at SUM is 98 ppt (428 to 526 ppt), which is modelled relatively poorly by TOMCAT$_{CON}$ at 54.8 ppt and improved by TOMCAT$_{OCS}$ to 60.1 ppt. A significant difference between SUM and the other two locations is that the topography in

the grid boxes for NWR and MLO is very spatially variable; for example, MLO is a high volcano on a relatively small island in the Pacific Ocean. These results suggest the model underestimates OCS concentrations around 1–3 km above sea level, and there is modest improvement between $TOMCAT_{OCS}$ and $TOMCAT_{CON}$.

The two NH tropical sites, MLO and KUM, exhibit a seasonal cycle in the measurements similar to that of NH continental sites, with slightly different phasing and a reduced seasonal amplitude, which is due to the influence of oceanic processes. Conversely, SMO, in the SH tropics, is more dominated by ocean processes and peaks earlier in the year. $TOMCAT_{OCS}$ at MLO, KUM and SMO shows varying levels of agreement with the observations. The RMSE in Fig. 1 at KUM is reduced by 56.8 %, and the SCA is improved compared to $TOMCAT_{CON}$ by 81 % from 17.7 to 3.4 ppt. MLO and SMO average concentration is better represented by $TOMCAT_{OCS}$, observing an RMSE improvement of −45.8 % and −30.2 %, respectively. Also note that the seasonality in Fig. 1 is out of phase with the observations, peaking approximately 1 month too late, while KUM is 2 months late. A challenge in diagnosing the misalignment in the Hawaiian sites is their proximity to the ocean, as the $2.8° \times 2.8°$ grid box is dominated by oceanic flux (as are all the boxes around it).

All four SH sites, SMO, CGO, PSA and SPO, show lower SCA in the observations than all NH sites, ranging from 25 ppt at SMO to 45 ppt at PSA, which is 80 % and 65 % less variation, respectively, than the forested site HFM. This emphasises the impact vegetation has on NH OCS seasonal cycle. Unlike SMO, the latter three sites are dominated much more by oceanic fluxes peaking in SH summer due to the association of phytoplankton growth with OCS emissions driven by solar radiation. The average concentration of OCS is underestimated by $TOMCAT_{OCS}$ and $TOMCAT_{CON}$ at CGO, PSA and SPO. Seasonal amplitude is overestimated in all models for all three sites. $TOMCAT_{OCS}$ overestimates the SCA values at CGO, SPO and PSA compared to the flask measurements by 5.1, 6.1 and 33.6 ppt, respectively. In contrast, $TOMCAT_{CON}$ overestimates these values by 18.2, 43.2 and 11.8 ppt, respectively. This suggests the reduction in Southern Ocean emissions in $TOMCAT_{OCS}$ adequately improves seasonality in OCS but could be reduced further. $TOMCAT_{SOTA}$ shows a much larger overestimation in seasonality.

## 4.2 Spatial distribution of modelled OCS compared to satellite observations

Figure 3 shows the spatial distribution of atmospheric OCS obtained by averaging ACE profiles across all longitudes and in 5° latitude bins (central column), along with the $TOMCAT_{OCS}$ profiles averaged in the same way (left column). The difference between the two datasets is shown in the right column ($TOMCAT_{OCS}$ minus ACE). ACE-FTS is

capable of measuring at altitudes between 6.5 and 30.5 km depending on latitude. Figure 3 shows that ACE tropospheric OCS mixing ratios (middle column) range from 425 to 500 ppt, peaking in the upper troposphere–lower stratosphere (UTLS) region, which extends from about 7 km in the NH ML and up to 17 km in the tropics. OCS values decline above and below the UTLS due to removal by vegetation and soil uptake at the surface and photochemistry in the stratosphere, leaving a peak in between that is significantly more prevalent in March–May (MAM) and June–August (JJA). As there is relatively little photosynthesis in the December–February (DJF) and MAM periods, OCS builds up in the atmosphere, followed by net removal throughout JJA and September–November (SON). Despite NH photosynthesis beginning slightly before JJA, there is a clear lag in removing OCS from the upper troposphere. The seasonal peak in OCS in the UTLS region only fully disappears in SON, suggesting there is roughly a 3-month delay in the influence of surface processes on the UTLS ambient mixing ratio. While this fluctuation is driven by seasonality in photosynthesis, the OCS peak is particularly large and extends lower in the atmosphere in the NH ML region, co-located with regions of especially large year-round anthropogenic emissions.

The tropopause height is captured adequately by $TOMCAT_{OCS}$ (which is forced by ERA-Interim) and is visible in the homogeneity of the difference around the UTLS. $TOMCAT_{OCS}$ agrees with ACE to within 25 ppt throughout most of the troposphere, which is about 5 % of the average estimated atmospheric value of OCS (484 ppt) (Montzka et al., 2007). Similar to the seasonal pattern visible in ACE, the tropospheric OCS mixing ratio in $TOMCAT_{OCS}$ peaks before the NH growing season (MAM). The maximum OCS concentration in $TOMCAT_{OCS}$ can be seen below 6 km (around 50–70° N) and peaks in MAM, is larger than maximum OCS observed by ACE, and persists throughout most of the year. The overestimation in the NH ML is broadly contained within a discrepancy of 25–50 ppt from ACE, with the exception of a few anomalies in MAM and JJA, potentially attributed to underestimated or slower surface OCS uptake or overestimated anthropogenic emissions in this region. The rate of removal of OCS is not quick enough in JJA to match the measurements exactly. This positive bias in the model below 10 km in the SH throughout DJF and MAM is probably unrelated to the NH positive model bias and would likely be resolved by weaker oceanic emission in the SH, despite already having been reduced by 40 % (Suntharalingam et al., 2008).

Differences between the model and observations in the stratosphere are broadly similar to those in the troposphere and are within ±25 ppt. However, there is considerable model underestimation at 24.5–30.5 km between 0 and 30° S of up to 65 ppt. This region shows a mean seasonal underestimation of between 24.6 % and 18.6 %, with a peak difference in JJA of 47.5 % around 29.5 km at 10° S. As this feature does not follow the pattern of the inter-tropical convergence zone

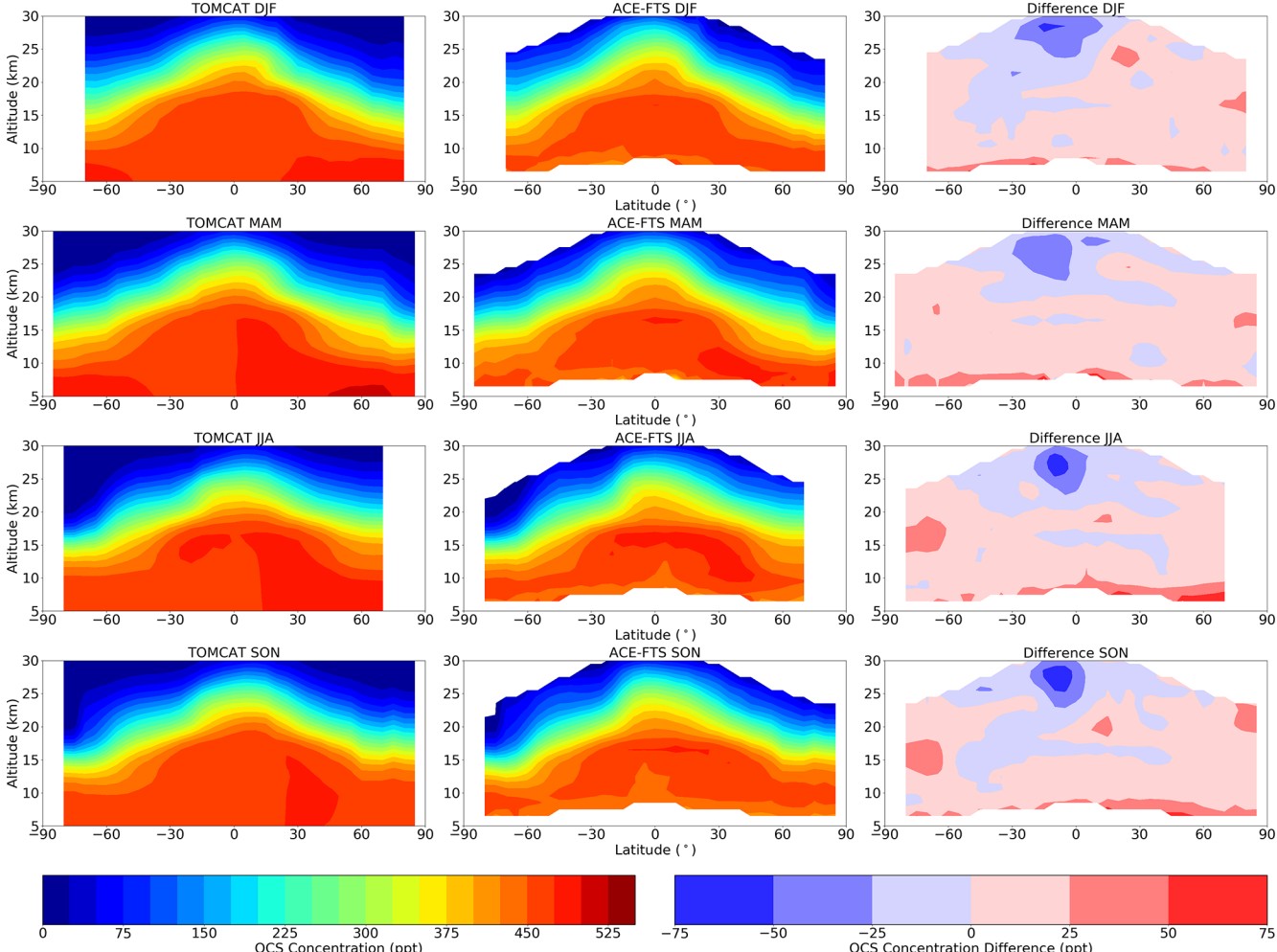

**Figure 3.** Seasonal zonal mean concentration (mixing ratio) of OCS (ppt) from TOMCAT$_{OCS}$ (left) and ACE (centre) and the difference between the two (TOMCAT$_{OCS}$ minus ACE, right) for the period of 2004 to 2018. TOMCAT$_{OCS}$ and ACE data averaged in 5° latitude bins and over all longitudes.

shifting with the hemispheric summertime period, it is unlikely that vertical fluxes are underestimated. The declining gradient in the stratosphere is steeper in the model than in ACE, which suggests that more OCS is being destroyed via photochemical processes in the model than in reality, which we examine in Sect. 4.3.

Figure 4 shows TOMCAT$_{OCS}$ profiles (blue) in 30° latitude seasonal bins compared to ACE (red), including the standard deviation (shown as error bars) of ACE at each altitude. When it is compared to observed OCS profiles from ACE-FTS, it is clear the model replicates the vertical structure of OCS. The negative discrepancy in the SH tropical stratosphere, visible in Fig. 3 and discussed above, can be seen most clearly in the third row of Fig. 4, as TOMCAT$_{OCS}$ deviates from ACE from 20 up to 30 km. However, as it remains within a standard deviation of ACE throughout the entire profile, this suggests the upper atmospheric sinks are modelled moderately well by TOMCAT$_{OCS}$. This applies to

most of the profiles compared in Fig. 4, in that the modelled TOMCAT$_{OCS}$ profiles generally remain within a standard deviation of ACE. The positive model biases in both hemispheres below 10 km in Fig. 3 can be seen in Fig. 4, such that the trend at these altitudes in TOMCAT$_{OCS}$ generally does not match ACE. Between 30° S and 90° N, ACE shows a depletion in OCS towards the surface from as high as 15 km driven by surface uptake. Here a more neutral or increasing gradient between 90 and 30° S is seen, as there is minimal vegetative uptake and a seasonal cycle strongly influenced by oceanic emission in this region (see Fig. 1).

## 4.3 Modelled OCS using reduced photochemical loss

The TOMCAT$_{OCS}$ model setup described in Sect. 3.1 and 3.3 is used for this experiment in which just the year 2010 is run with an adjusted atmospheric photolysis rate of 75 %. The intention of this simulation is to make a simple preliminary

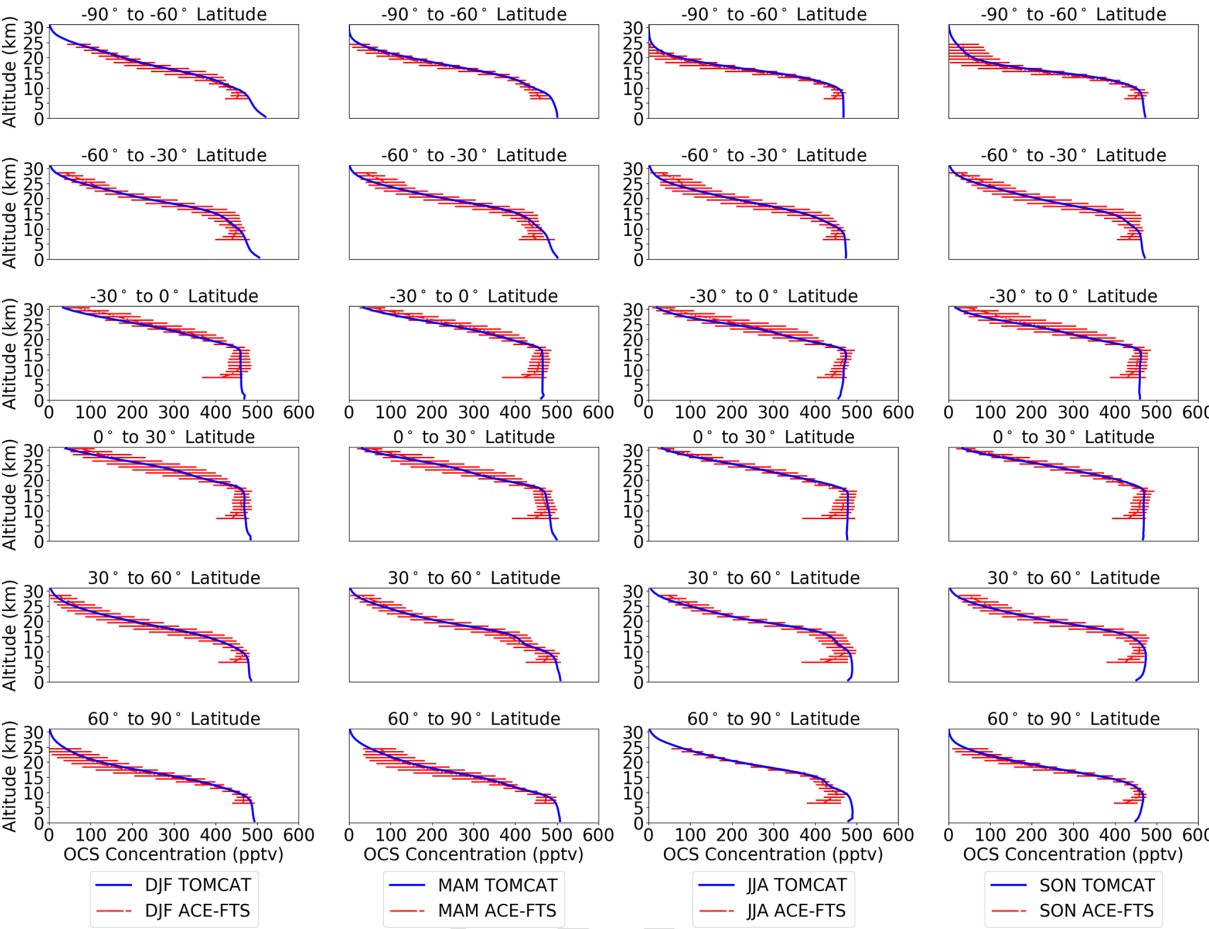

**Figure 4.** Seasonal mean vertical profiles of OCS concentration (mixing ratio, ppt) from TOMCAT model output (blue) and ACE (red) for six different latitude regions. The error bars are standard deviation of ACE at each altitude level. All profiles are seasonal averages between the years 2004 and 2018.

assessment of if photolysis alone will correct the underestimation in the SH stratosphere. Figure 5 shows the modified version of TOMCAT$_{OCS}$ on the left, ACE measurements made in 2010 in the middle and the difference between the two on the left (TOMCAT$_{OCS}$ minus ACE). When compared with Fig. 3, the reduced removal of OCS improves the differences between the model and measurements above approximately 20 km, which is to be expected, as this is generally where photolysis is active. The regions that exceed a difference of ±25 ppt are limited to isolated pockets throughout the year that noticeably occur more in SON, in the tropics around 20–30 km and in the SH around the tropopause. Using only a year of data removes a lot of the smoothing we see in Fig. 3, which accounts for some of the differences in Fig. 5. However, there are similar features between the two, specifically persistent (but slightly reduced) underestimation in the model in the tropical SH. We also see an increase in positive model bias, most obviously in the NH between 0 and 30 ° N in DJF and SON. Overall, we find that differences between the model and measurements were merely shifted by

a 25 % reduction in photolysis, introducing biases elsewhere. Further testing, including a simulation using a photolysis rate reduced by 50 %, exacerbates these differences further (see Fig. S11), leading to the conclusion that other processes, potentially including transport or convection, require correction to fully resolve these issues in the model.

## 5   Discussion

After a 10-year spin-up period, the TOMCAT$_{OCS}$ simulations of atmospheric OCS concentrations and the vegetative flux, which are dependent on one another in the model, are in equilibrium between 2004 and 2018. By utilising the LRU-GPP approach, we estimate a mean yearly vegetative OCS uptake of 629 Gg S yr$^{-1}$, which is within the range and uncertainty of the magnitude of this flux from previous top-down studies (see Table 2). Our estimate is also in the range of recent bottom-up estimates by Kooijmans et al. (2021), Maignan et al. (2021) and Abadie et al. (2022) (576–756 Gg S yr$^{-1}$). Our total vegetative and soil

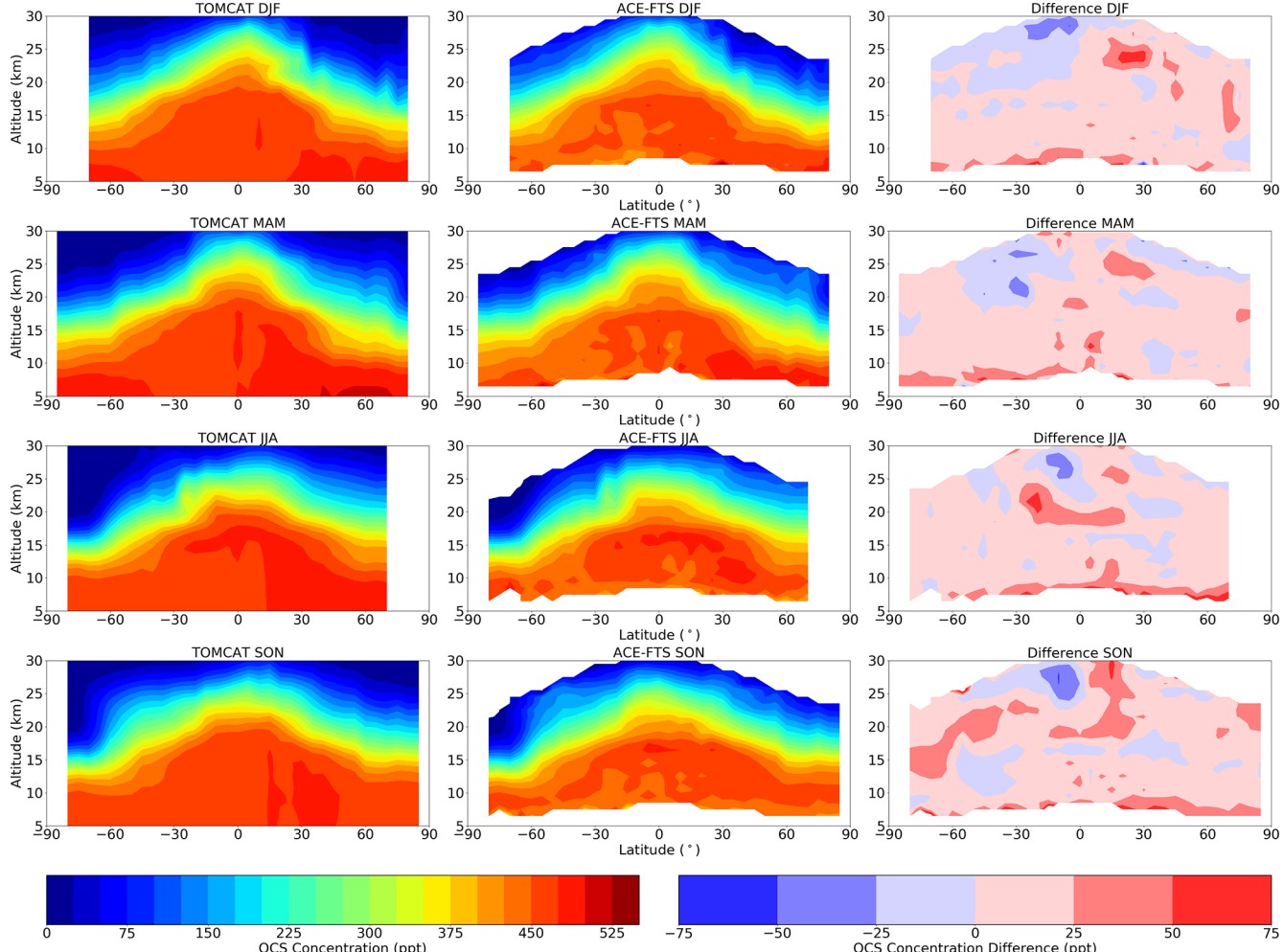

**Figure 5.** Seasonal zonal mean concentration (mixing ratio) of OCS (ppt) from TOMCAT$_{OCS}$ (left) only for the year 2010 and with a photolysis rate 0.75 times that of TOMCAT$_{OCS}$, from ACE (centre), and the difference between the two (TOMCAT$_{OCS}$ minus ACE, right) for the period of 2004 to 2018. TOMCAT$_{OCS}$ and ACE data are averaged in 5° latitude bins and over all longitudes.

sinks agree with findings from Berry et al. (2013) and inversion studies by Kuai et al. (2015), Ma et al. (2021) and Remaud et al. (2022) and are at approximately $\pm 150\,\text{Gg S yr}^{-1}$ (15 %). We balance the OCS budget by implementing an enlarged oceanic $CS_2$ emission source, for the exclusive reason that it is focused over the tropics (Kettle et al., 2002) rather than a flux originating from oxidised $CS_2$. Bottom-up estimates recommend a constraint on global oceanic emission of OCS to approximately $285–350\,\text{Gg S yr}^{-1}$ (Lennartz et al., 2017, 2020, 2021), significantly lower than the fluxes required to balance our budget and thus bringing our tropical ocean estimate into question. It is clear that tropical fluxes are still uncertain; however, inverse modelling of OCS fluxes shows that some combination of a larger tropical oceanic source and vegetative sink resolves the budget and produces adequate model comparison with independent observations (Ma et al., 2021; Remaud et al., 2022).

TOMCAT$_{OCS}$ output agrees with ACE-FTS profiles of OCS within 25 ppt throughout the majority of the observed atmosphere (approximately 5–30 km), suggesting the sinks in the upper atmosphere are modelled well, with the exception of some discrepancies in the lower troposphere and the tropical lower stratosphere. Photochemical destruction is important in our understanding of atmospheric OCS, and due to photolysis in the stratosphere, the model displays a declining vertical gradient above the tropopause. Our total estimate for this flux is $154\,\text{Gg S yr}^{-1}$, an upward revision of about 40 % compared to the previous work of Kettle et al. (2002) and larger than all other estimates in Table 2. Comparison of TOMCAT$_{OCS}$ with ACE profiles shows a good representation of the free troposphere, suggesting that we have found a suitable balance of fluxes at the surface, the spatial variability of which requires improvement. Overestimation in the NH ML region in JJA and SON suggests that surface emissions could be overestimated or that surface uptake does not

initialise quickly enough or strongly enough to remove OCS from the atmosphere at the start of the growing season.

To assess the OCS surface seasonality modelled by $TOMCAT_{OCS}$ we compare it to two other simulations: $TOMCAT_{CON}$ and $TOMCAT_{SOTA}$. The vegetation, soil and ocean emission fields used to drive these models can be found in the Supplement. When compared with surface flask observations, we show that the OCS budget used to calculate $TOMCAT_{OCS}$ reduces RMSE compared to the control, $TOMCAT_{CON}$, at most sites by approximately 25 % and as much as 57 % at KUM but degrades some RMSE values, notably that at MHD (see Fig. 1). We also show improvements in RMSE at NH continental sites, especially the forested sites of LEF and HFM, but there is still moderate underestimation in NH vegetative uptake. Comparing RMSE in the monthly anomalies between $TOMCAT_{OCS}$ and $TOMCAT_{CON}$ (see Fig. 2) shows that improved average concentration contributed significantly to improving RMSE in Fig. 1, as $TOMCAT_{OCS}$ improves SCA by only 5 % compared to $TOMCAT_{CON}$.

The Hawaiian sites, MLO and KUM, show significantly improved RMSE with $TOMCAT_{OCS}$, and some improvement in SCA and phasing, suggesting an enhancement in tropical oceanic emission is reasonable. The lack of OCS measurements in the tropics poses a challenge to both quantifying surface OCS exchange in this region both from a mechanistic perspective and from constraining inverted fluxes (Whelan et al., 2018; Ma et al., 2021; Remaud et al., 2022). While $TOMCAT_{OCS}$ shows an adequate comparison with tropical surface sites and a vertical comparison within the variability of ACE, we acknowledge that no attempt has been made in this work to experiment with reducing tropical surface OCS uptake, which has been suggested as an alternative solution to balance the OCS budget (Ma et al., 2021). Overestimation in $TOMCAT_{OCS}$ SCA at SH sites CGO, PSA and SPO indicates a reduction in oceanic emissions in this region is necessary due to the limited continental landmass and associated uptake.

The method of estimating vegetative uptake using the LRU approach does have limitations, such as calculating OCS uptake using a constant LRU value of 1.6, which is not representative of reality. LRU values have shown to vary from approximately 1.0 to 4.0 based on different plant type and atmospheric conditions (Sandoval-Soto et al., 2005; Seibt et al., 2010; Stimler et al., 2010, 2012; Kooijmans et al., 2019). Our estimation of vegetative uptake in this work does not replicate OCS uptake universally, and it is unclear if this is due to localised differences in LRU or in the GPP fields themselves. When compared to bottom-up vegetation uptake estimates from the ORCHIDEE model used to drive $TOMCAT_{SOTA}$ (Fig. S8), the LRU approach shows similar spatial distribution, magnitude and seasonality (Fig. S5). There are notable differences in tropical locations in terms of magnitude year-round, i.e. South America in January and Africa in April. These regions should have a low impact on

seasonality in the mid-latitudes and high latitudes in the NH. However, despite the similarities in the vegetation fields between the two methods, Fig. 2 still shows considerable differences in seasonality in NH sites. The 2-month delayed phasing at BRW is observed by Remaud et al. (2023) in their inter-model comparison study that employs state-of-the-art bottom-up fluxes, similar to those used in this work, and in some cases their results are the same as ours. They attribute this difference to overestimation in NH ocean sources and/or underestimation in vegetative uptake. This agrees with the results of inversion studies by Ma et al. (2021) and Remaud et al. (2022) as both inversions show increased uptake and decreased emissions of OCS in the NH mid-latitudes to high latitudes in their posterior fluxes. However, we still lack an explanation for the poor seasonality at the forested sites of LEF and HFM in $TOMCAT_{SOTA}$ (underestimated by approximately 100 ppt at both sites vs. the SCA in NOAA measurements). Modelling of OCS and comparing a mechanistic approach and the LRU method in ORCHIDEE done by Maignan et al. (2021) both show that they should behave similarly on a global and seasonal scale. However, further work is required to better understand the relationship between OCS uptake and GPP and why the fluxes driving $TOMCAT_{SOTA}$ do not capture the seasonality of OCS surface mixing ratios.

While soil uptake has been scaled appropriately according to the literature, the distribution is based on work by Kettle et al. (2002) and has since been updated, e.g. in Abadie et al. (2022). Comparing soil uptake used in $TOMCAT_{OCS}$ to that in $TOMCAT_{SOTA}$ (Fig. S6 vs. Fig. S9), where the latter is estimated using ORCHIDEE, we see considerable differences that partly account for the different seasonal cycles in the NH, particularly at ALT, SUM, BRW and MHD. Figure S9 shows a reasonably homogeneous distribution and seasonality compared to Fig. S6, which shows far more annual variability and spatial variation.

## 6 Conclusions

A 3-D chemical transport model was used to compare three OCS flux scenarios: one utilising the LRU approach to quantify vegetative uptake and a series of scaled fluxes ($TOMCAT_{OCS}$) and two using bottom-up fluxes originating from the literature ($TOMCAT_{CON}$ and $TOMCAT_{SOTA}$). $TOMCAT_{CON}$ uses fluxes estimated by Kettle et al. (2002), and $TOMCAT_{SOTA}$ uses a series of novel fluxes from recent literature (Lennartz et al., 2017; Zumkehr et al., 2018; Stinecipher et al., 2019; Lennartz et al., 2020; Maignan et al., 2021). All simulations are compared with surface anomalies from the NOAA-ESRL flask network, and $TOMCAT_{OCS}$ is compared to ACE-FTS satellite observations. This study is novel in the extended time period analysed and the quality of vertical comparison with the most recently available ACE-FTS satellite measurements (version 4.1). Furthermore, we see good comparisons with ACE-FTS throughout most of the

atmosphere, which suggests the free troposphere and gradient above the UTLS is well represented by TOMCAT$_{OCS}$. Therefore, there is a suitable balance between model sources and sinks from the surface to simulate atmospheric OCS. Future applications of the TOMCAT$_{OCS}$ model could encompass its use in comparison with and interpretation of nadir-viewing satellite observations, e.g. from the Infrared Sounding Interferometer (IASI) instruments.

TOMCAT$_{OCS}$ and TOMCAT$_{CON}$ surface concentration is compared, and the former is shown to reduce RMSE compared to 14 NOAA-ESRL by 12.5 % across all 14 sites, up to 20 % when neglecting MHD. Further, surface anomalies (monthly mean minus annual mean) are compared between all three model runs, yielding a RMSE that removes annual mean and focuses solely on seasonality. TOMCAT$_{OCS}$ reduces the RMSE in the anomalies by 18.7 % and 52.4 % compared to TOMCAT$_{CON}$ and TOMCAT$_{SOTA}$, respectively. Adequately modelling seasonality globally proved to be a challenge, and while TOMCAT$_{OCS}$ performed the best relative to NOAA-ESRL flask observations, the seasonal cycle amplitude was misaligned by a mean absolute amount of 26.6 ppt by TOMCAT$_{OCS}$ and by 43.7 ppt by TOMCAT$_{SOTA}$. We have shown that the LRU approach for quantifying vegetative uptake yields similar annual estimates (629 Gg S yr$^{-1}$) to mechanistic and inversion approaches (657–756 Gg S yr$^{-1}$) and resembles spatial variability (Maignan et al., 2021; Remaud et al., 2022). To suitably estimate a total biosphere uptake that reflects recent inversion studies of roughly 893–1053 Gg S yr$^{-1}$ (Remaud et al., 2022; Ma et al., 2021), soil uptake is uplifted by 2.5 times, yielding a combined total of 951 Gg S yr$^{-1}$. To bring the budget into balance we increase total net oceanic emissions to 650 Gg S yr$^{-1}$ from the starting point of 279 Gg S yr$^{-1}$ (Kettle et al., 2002). Overall, we draw similar conclusions to other works that the tropics are a likely location for a compensatory source of OCS.

Here, we make recommendations for advancing this work. The following changes are necessary in the future to improve the GPP-LRU approach, such as using inter-annually varying GPP and $CO_2$ mixing ratios and a temporally and spatially resolved LRU. It is challenging to achieve using a high-resolution LRU product on a global scale; however, plant-functional-type-dependent datasets of LRU are available (Seibt et al., 2010; Whelan et al., 2018; Maignan et al., 2021). Hence, an initial step would be just to vary LRU based on ecosystem on a continental or ecosystem scale. Advances are being made in this area, with mechanistic and LRU approaches emerging that reduce uncertainty in OCS vegetative uptake (Kooijmans et al., 2021; Maignan et al., 2021). The use of an enhanced tropical ocean source to balance the budget is justified in this work by offering suitable satellite and surface observational comparisons. However, we acknowledge that oceanic emissions alone may not account for this discrepancy, and this is not a perfect solution for balancing the global OCS budget based on the performance of TOMCAT$_{OCS}$ at MLO and KUM.

While optimised fluxes from inversion studies show the most up-to-date distribution of OCS fluxes at the surface, what they lack is information on bottom-up processes. As we have seen in Fig. 2, TOMCAT$_{SOTA}$ lacks suitable seasonality in terms of both SCA and phasing, for many NH surface measurement sites. More work is required to dissect the impact of individual fluxes on seasonality and further understand why bottom-up fluxes differ so greatly from posterior inverted fluxes, most importantly vegetative uptake and oceanic emissions.

While we have shown that TOMCAT$_{OCS}$ compares well with satellite observations, the region between the surface and approximately 6 km, which is not measured by ACE-FTS, could hold a lot of information useful in resolving surface fluxes. Measurements at the surface are sensitive to minor flux changes, although in the well-mixed middle- to upper troposphere these spatial changes are less important. Validation of model output to ground-based Fourier transform spectrometer column OCS measurements could improve our understanding and ability to model the lower troposphere. Furthermore, incorporating measurements with vertical information into an inversion scheme has been shown, specifically when using HIPPO flight data, to improve the posterior OCS fluxes (Ma et al., 2021). Therefore, further study following on from this work will be aimed at deriving an a posteriori set of fluxes using an inversion scheme based on an up-to-date prior that uses surface observations and a dataset containing vertical information near the surface.

**Code and data availability.** Anthropogenic OCS emission data are available at https://portal.nersc.gov/project/m2319/ (Campbell, 2022; Zumkehr et al., 2018). The GPP dataset is available at https://doi.org/10.7488/ds/1461 (Slevin et al., 2016). ACE-FTS data are available at http://www.ace.uwaterloo.ca/data.php (ACE-FTS, 2022). NOAA-ESRL surface flask measurements of OCS are available at https://www.esrl.noaa.gov/gmd/dv/data/ (NOAA Global Monitoring Laboratory, 2022). Model data are available at https://doi.org/10.5281/zenodo.6368542 (Cartwright, 2022).

**Supplement.** The supplement related to this article is available online at: https://doi.org/10.5194/acp-23-1-2023-supplement.

**Author contributions.** Model runs and data analyses were performed by MPCa with support from RJP. JJH and MPCh designed the study. CW provided the $CO_2$ model data. Control OCS emissions were provided by PS. The TOMCAT model is maintained and updated by the team at the University of Leeds (MPCh, WF, CW and RJP). The manuscript was written by MPCa with contributions from all co-authors.

**Competing interests.** The contact author has declared that none of the authors has any competing interests.

**Disclaimer.** Publisher's note: Copernicus Publications remains neutral with regard to jurisdictional claims in published maps and institutional affiliations.

**Acknowledgements.** Computation and data analyses were carried out on the ARC computing system at Leeds and the ALICE/SPECTRE system at Leicester. The authors thank Peter Bernath for access to ACE-FTS satellite observations; Darren Slevin, Simon Tett, and Mat Williams for access to GPP data from JULES; and Stephen Montzka and James Elkins and other contributors from NOAA for providing the flask measurements. The authors thank Marine Remaud, Jin Ma, Maarten Krol and all other principal contributors to the TRANSCOM inter-model comparison study for providing gridded fluxes used in the TOMCATSOTA simulations.

Michael P. Cartwright thanks the University of Leicester for providing a studentship and the National Centre for Earth Observation for additional funding to undertake this work

**Financial support.** This research has been supported by the UK Research and Innovation Natural Environment Research Council (grant no. PR140015), the University of Leicester (Leicester institute of Space and Earth Observation Studentship), and a CASE award from the National Centre for Earth Observation.

**Review statement.** This paper was edited by Anita Ganesan and reviewed by Maarten Krol and one anonymous referee.

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

## Remarks from the typesetter