# Peer review of "Constraining the budget of atmospheric carbonyl sulfide using a 3-D chemical transport model"

_Atmospheric Chemistry and Physics, 2022_

## Referee Comment (RC2)

[referee-annotated manuscript omitted]

---

## Author Comment (AC1)

**Constraining the budget of atmospheric carbonyl sulfide using a 3-D chemical transport model**

**Final Response to Referee Comments**

Due to the extensive corrections required, large portions of the text are now different. Hence some referee comments are now unnecessary. We have tried to address every comment as best we can, including a line number where it best relates in the current text. Author responses can be found in italics.

**Commenter Summaries**

**First**

Michael P. Cartwright and co-authors present an evaluation of two OCS flux inventories using OCS concentration observations from 14 NOAA towers and from the ACE-FTS satellite instrument. The first inventory is mainly based on Kettle et al. (2002) and is used as a control for the second inventory. The authors define this second inventory using the LRU approach to compute vegetation OCS fluxes, then scaling the other OCS flux components to obtain a balanced global OCS budget. The OCS fluxes from both inventories are transported with the TOMCAT atmospheric transport model. Finally, the surface OCS flux components are evaluated by comparing the seasonal cycles of the simulated OCS concentrations to NOAA flaks measurements. The vertical profiles of the simulated OCS concentrations are also evaluated against ACE-FTS OCS concentration profiles between 5 and 30 km. The authors conclude on a better performance of the new balanced OCS flux inventory compared to the control simulation in terms of seasonal cycle representation. The strength of this work is to make use of ACE-FTS observations to evaluate the simulated OCS concentration vertical profiles. However, major revisions should be considered before publication.

**Second**

This paper presents forward simulation of OCS using the TOMCAT model. Two main simulations are presented: a control simulation and a simulation in which GPP from the JULES model is converted into an OCS flux using the LRU approach. Results generally show an improvement with the surface observation network, and a favourable comparison with the ACE-FTS observations. The paper is well written, with a clear structure. However, the results are rather thin in the sense that the field of OCS research is moving rapidly, and formal inversion systems are now in place. In that sense, the "hand-adjusted-flux" approach in this paper may be a bit outdated. The comparison with ACE-FTS and the use of a new model (TOMCAT) and biosphere model (JULES GPP) provides sufficient new information. The paper  might provide a valuable addition to the existing literature, after addressing some major issues.

**Major Comments**

**First**

1. This work makes insufficient use of OCS state-of-the-art. References to recent studies are missing in the introduction. For example, when describing atmospheric OCS trend, Hannigan et al. (2022) should be mentioned as they found positive trends in the troposphere and in the stratosphere

between 2008 and 2016 at most of the studied sites. The reference of the study by Glatthor et al. (2017) should also be contrasted with the increasing trend found by Hannigan et al. (2022) in the free troposphere at the Jungfraujoch site between 2008 and 2016, followed by a decreasing trend since 2016–2017. Ma et al. (2021) should be presented as an inversion study and completed with Remaud et al. (2022) that also conclude to a missing source in the tropics. The underestimated OCS anthropogenic source suggested by Zumkehr et al. (2018) was also supported by Aydin et al. (2020).

Using the inventory of Kettle et al. (2002) is also a major weakness as many studies have provided new OCS flux estimates since. Therefore, the conclusion that TOMCATocs gives better results than TOMCATcon does not seem relevant when TOMCATcon is based on out-of-date estimates (except for the anthropogenic emissions from Zumkehr et al., 2018). Many limitations that were highlighted in OCS literature arise from using the inventory from Kettle et al. (2002). For example, oxic soil contribution considers a constant atmospheric OCS concentration while recent studies have shown the importance of considering variable atmospheric concentrations for both soil and vegetation OCS fluxes (Kooijmans et al., 2021, Maignan et al., 2021, Abadie et al., 2022). Important processes are also not included in the control inventory. Indeed, oxic soils can not only take up OCS but also produce OCS, and it has recently been shown that anoxic soils cannot be neglected at the global scale (Abadie et al., 2022).

In Section 3.3, why choosing to use a single constant LRU value while several studies provide PFT-dependent LRU values? For example, these sets of LRU per PFT can be found in Seibt et al. (2010), Whelan et al. (2018), Maignan et al. (2021). Moreover, as JULES land surface model distinguishes several PFT categories, it would be possible to use PFT-dependent LRU values.

In Section 3.4, the OCS fluxes that were adjusted to obtain a balanced budget should be contrasted with more recent estimates. For example, choosing to scale CS2 oceanic emissions to 439 GgS/y is not supported by Lennartz et al. (2020) who estimated a total source of 70 GgS/y from CS2. An oxic soil budget of 322 GgS/y is also not in line with the recent estimates of Kooijmans et al. (2021) and Abadie et al. (2022) based on the mechanistic soil model of Ogée et al. (2016).

More recent studies should be added in Table 2 to compare to the OCS budget from this work, such as Maignan et al. (2021), Kooijmans et al. (2021), Remaud et al. (2022).

2. The scaling of OCS fluxes to better match estimates made after Kettle et al. (2002) and to obtain a balanced OCS budget seems quite arbitrary. Such adjustments should be made using an inversion framework as done in Ma et al. (2021) or in Remaud et al. (2022). Without an analytical inverse system that optimizes the fluxes, why aiming at a balanced COS budget? A balanced OCS budget is also not required if analyzing the detrended OCS concentrations.

Moreover, this scaling assumes that the OCS flux spatial distribution of each component is not modified compared to the control inventory, which might not agree with flux distribution obtained in more recent studies.

3. Not considering interannual variations is a strong assumption that should at least be better justified. The study from Chen et al. (2017) does not conclude that interannual variability in GPP amplitude can be neglected. GPP interannual variability could easily be included in this work as GPP is modeled by JULES. Considering only the year 2010 does not reflect the yearly increase in atmospheric CO2 concentration and the fertilization effect.

Otherwise, could the impact of not considering OCS flux interannual variability be quantified? For example, OCS vegetation uptake could be defined as a first order relationship with OCS mixing ratio. Therefore, inter-annual variations in OCS vegetation flux might have a strong impact on the simulated atmospheric OCS concentrations.

4. It is not clear what the goal of this study is, and the title is confusing. It is mentioned that this work evaluates the suitability of gross primary productivity to estimate the OCS vegetative uptake. However, doing so by comparing a vegetation OCS uptake based on GPP to a vegetation OCS uptake based on NPP and NDVI seems outdated (Sandoval-Soto and Stanimirov, 2005). Moreover, the modification of several OCS flux components with the rescaling makes it difficult to compare seasonal cycles of TOMCATcon and TOMCATocs regarding vegetation OCS fluxes. Please specify clearly what are the main goals of this study.

Should this work focus more on the advantage of using ACE-FTS compared to other available OCS concentration observations? Or on the information that can be retrieved from ACE-FTS about the modelling of OCS atmospheric sinks?

**Second**

This paper presents forward simulation of OCS using the TOMCAT model. Two main simulations are presented: a control simulation and a simulation in which GPP from the JULES model is converted into an OCS flux using the LRU approach.

Results generally show an improvement with the surface observation network, and a favorable comparison with the ACE-FTS observations.

The paper is well written, with a clear structure. However, the results are rather thin in the sense that the field of OCS research is moving rapidly, and formal inversion systems are now in place. In that sense, the "hand-adjusted-flux" approach in this paper may be a bit outdated. The comparison with ACE-FTP and the use of a new model (TOMCAT) and biosphere model (JULES GPP) provides sufficient new information. The paper can might provide a valuable addition to the existing literature, after addressing some major issues.

First, when the results are described in the paper, often hand-waving argumentation is used to explain the deviations between model and observations. "Likely caused", "could be attributed". Here, we have to believe the judgement of the authors, since rarely additional arguments are presented. Likewise, the underestimation of modelled OCS in the tropical stratosphere is explained by too fast removal. These observations call for additional simulations to verify whether speculations hold true. Two suggestions here: (1) a simulation with tagged tracers to be used in a more detailed analysis of e.g. seasonal cycles (2) a simulation with reduced photochemical removal in the tropical stratosphere. Presentation of the results would give the paper more body.

Second, the LRU approach is defendable, and uses seasonal $CO_2$ mixing ratios to convert GPP into a OCS flux. It remains unclear what is taken for the OCS mixing ratios here. In the recent papers of Ma et al. (quoted), and Kooijmans et al. (Biogeosciences, 2021, not quoted) it is clearly shown that OCS fluxes become substantially smaller in regions with low OCS abundance ($F\_OCS = -vd*OCS$). I am a bit surprised that nothing is mentioned on how OCS mixing ratios are used to convert GPP to a OCS flux. If a constant value of e.g. 500 ppt is taken, the final OCS flux may become substantially smaller. Kooijmans et al., (2021) report a drop in SIB4 from 922 Gg S yr−1 in the original SiB4 to 753 Gg S yr−1 when accounting for varying OCS mixing ratios.

Finally, the field of OCS research is moving fast. The paper therefore misses quite some recent references that are relevant for the work. The authors should update the reference list (and discussions) with more recent papers.

Further comments are in the accompanying annotated pdf file.

**Minor Comments**

**RC1**
Abstract:

L25: "At the surface, the model captures background concentrations at most of the surface sites to within the maximum and minimum of the seasonal measurements". It does not seem to be a strong condition to satisfy. It might be better to highlight results on the seasonal cycle amplitude or the phase. – *The error bars represented in Figure 1 are now standard deviation, rather than maximum and minimum. Results of SCA are highlighted in the new abstract.*

1.Introduction:

L49: "The main source of atmospheric OCS is oceanic emission". This could be replaced by "one of the main sources" as anthropogenic emissions are also a major source of OCS. For example, Zumkher et al. (2018) estimated an anthropogenic OCS source of about 400 GgS/y. Aydin et al. (2020) suggested that this estimate could be underestimated with an anthropogenic OCS source of about 600 GgS/y. – *Updated. Line 64.*

L57: "with estimates ranging from 210 to 2400 Gg S yr-1 (Kettle et al., 2002; Sandoval-Soto and Stanimirov, 2005; Suntharalingam et al., 2008; Berry et al., 2013; Glatthor et al., 2015; Kuai et al., 2015; Launois et al., 2015b; Ma et al., 2021)". Add references to more recent studies such as Maignan et al. (2021), Kooijmans et al. (2021), Remaud et al. (2022) for vegetation OCS uptake estimates. – *Updated. Line 75-77.*

L61: "OCS hydrolysis also occurs in soil, again catalysed by carbonic anhydrase". Note that OCS can also be consumed by other enzymes in soils, such as nitrogenase, CO dehydrogenase, or CS2 hydrolase (Smith and Ferry, 2000; Masaki et al., 2021). – *Updated. Line 81-82.*

L63: "with an estimated annual loss of 127-355 Gg S (Kettle et al., 2002; Montzka et al., 2007; Berry et al., 2013; Glatthor et al., 2015; Kuai et al., 2015)". More recent studies should be mentioned, such as Kooijmans et al. (2021) and Abadie et al. (2022) that lead to smaller soil OCS budgets. – *Updated. Line 83.*

L65: "Soil has also been observed to act as an emitter of OCS in warm conditions (Maseyk et al., 2014)". This was not observed only for warm conditions. OCS emissions have also been related to soil types (Whelan et al., 2013), nitrogen content, light radiations reaching the soil surface (Spielmann et al., 2019; Kitz et al., 2020). – *Updated. Line 85-88.*

L69: "the latter of which has been used as a benchmark for more recent studies". Please add the references of the studies. – *Updated. Line 92-93.*

2. Observations:

Section 2.1: Please provide and detail the uncertainties associated with ACE-FTS retrievals. – *Updated. Line 145-147.*

3. Chemical transport modelling of OCS:

Section 3.1: What is the timestep used to run the TOMCAT model? – *6 hours. Updated. Line 172-173.*

L139 to L224: Please make it clearer which fluxes have been used for TOMCATcon and which one have been used for TOMCATocs. – *Updated. TOMCAT_{CON} uses inventory in Section 3.2 and TOMCAT_{OCS} uses inventory in Section 3.3.*

L153: "The three sink terms are an oceanic sink, soil uptake and a vegetative sink". OCS photolysis in the stratosphere and OCS oxidation by OH radical in the troposphere should also be included in OCS sinks, as atmospheric OCS reactions are not explained before in section 3.1. – *We include a summary of OH and Photolysis loss in Section 3.1, before describing all the fluxes. It is clear the same scheme is used for all model runs. Line 166-170.*

Equation 1: Precise the unit for each term of this equation. What is used for OCS background concentration? – *Updated. Line 222-226.*

L173: Please replace "LRU is the normalised ratio of OCS assimilation rates to CO2 at the leaf-scale. This is then normalized by background concentrations of the two gases" by "LRU is the ratio of OCS assimilation rates to CO2 at the leaf-scale, both normalized by their respective concentration". – *Updated. Line 223-224.*

L180: "but is slightly under half that of the largest estimation of 1115 Gg S in Table 2 from Montzka et al. (2007)". Launois et al. (2015) estimated a larger plant OCS uptake than Montzka et al. (2007) for the ORCHIDEE land surface model. – *Updated. Line 242.*

L204: "at Northern Hemisphere (NH) NOAA-ESRL sites". Please precise which sites and whether they receive air masses mainly coming from the ocean. – *Updated. Line 277-278.*

Table 2: Why were more recent studies not included in this table for comparison? Such as Maignan et al. (2021), Kooijmans et al. (2021), Remaud et al. (2022). – *Updated to include Remaud et al. (2022).*

4. Results:

L243 to L245: "TOMCATCON was initialised using OCS values in each grid box from TOMCATOCS, after 10 years (1994 – 2003) spin-up. Only 2004 monthly mean mixing ratios from TOMCATCON have been included, as this flux inventory has a 245 net negative budget and therefore a negative trend over longer periods". Should this be in Section 3 as it is related to the method? – *Moved to section 3. Line 187.*

L244: "Only 2004 monthly mean mixing ratios from TOMCATCON have been included, as this flux inventory has a net negative budget and therefore a negative trend over longer periods". Please precise the net negative budget. The trend in atmospheric COS concentrations should not be an issue if you remove the trend and compare the detrended atmospheric concentrations. – *Updated. Line 188. Figure 2 now presents monthly anomalies.*

L249: "Error bars associated with the observations represent the maximum and minimum values for each month at every site". Representing the standard deviation would be a better indication of the uncertainty of the mean value. – *Updated. Figure 1 error bars are now represented by standard deviation.*

L253: "Comparisons between TOMCATocs and TOMCATcon are shown here to emphasise the improvements made by the flux inventory developed in this study". How could the improvements obtained with TOMCATocs on atmospheric OCS concentrations be compared to the improvements

made when using inversion systems such as in Ma et al. (2021) and Remaud et al. (2022)? – *We compare TOMCATOCS to a new model run TOMCATSOTA, which utilises fluxes that are more up-to-date in the literature and similar or the same as those used in as prior fluxes by Ma et al. (2021) and Remaud et al. (2022). See discussion and supplement.*

L254: "The root mean square error (RMSE) for the entire period is shown for each site, alongside the seasonal cycle amplitude (SCA)". Precise that in the following you also compare the phases of observed and simulated seasonal cycles. – *SCA is now displayed in Figure 2.*

L255: "Generally, there is an improvement in RMSE across all the sites, but in some cases, there is a degradation, which is mostly attributed to background concentration, rather than the model's ability to capture a suitable seasonal cycle, hence both are shown." By "background concentration", do you mean the average concentration? If so, could you please show that the degradation in RMSE is due to the average concentration? – *I did mean average concentration. Figure 1 shows TOMCAT$_{OCS}$ and TOMCAT$_{CON}$ monthly mean model values, but Figure 2 shows monthly anomalies. So we are able to distinguish the impact from poor estimation of average concentration and of the seasonality.*

L264: "This seasonal cycle resembles that of CO2, hence GPP is a suitable proxy for calculating OCS uptake". Please rephrase as similar seasonal cycle is not reason enough to use GPP as a proxy of vegetation OCS uptake. – *I agree. This was rephrased. Line 377.*

L275: "Here we show realistic amplitudes in the seasonal cycle from TOMCATOCS, 76 ppt at LEF and 71 ppt at HFM, compared to observed values of 123 ppt and 128 ppt, respectively". Please rephrase as the SCA at these two sites are still largely underestimated. – *Rephrased. Line 399-401.*

L279 to 282: The constant LRU value used in this study could be compared to other LRU estimates for the same vegetation types found at LEF and HFM to see if it could be underestimated. If a constant OCS mixing ratio was used to compute OCS vegetation uptake, this could also affect the SCA. – *I have updated the text in places to make it clear we are using OCS concentration from online within the model. A brief discussion of LRU in the literature is made in Section 5: Line 565-570.*

L329: "OCS values decline above and below the UTLS due to removal by photosynthesis at the surface". Soils can also absorb OCS at the surface. – *Updated. Line 456-457.*

L345: "potentially attributed to slower surface OCS uptake". It could also be due to an underestimated surface OCS uptake. – *Updated. Line 472-473.*

Figure 2: What could explain the net distinction between higher mixing ratios in NH compared to SH found in TOMCAT in JJA and SON? – *Weaker removal of OCS in the SH.*

L364: "this suggests the upper atmospheric sinks are modelled well by TOMCATocs". Isn't it in contradiction with the steeper gradient of TOMCATocs in the stratosphere mentioned above? – *Slightly rephrased to play down the performance. Line 492-493.*

Section 4.2 attributes model-observation mismatches to OCS sources or sinks, what about the potential mismatches from TOMCAT transport? – *This is mentioned in Section 4.3. Line 516.*

5. Discussion:

L376: "the TOMCATocs simulations of atmospheric OCS concentrations and the vegetative flux, which are dependent on one another in the model". The simulated OCS concentrations are dependent on vegetation OCS fluxes that were transported, but it is not said which OCS

concentrations are used to compute the vegetation OCS fluxes. – *OCS concentration from the model is used in the calculation of vegetative uptake. See Line 228.*

L385: "inverse modelling of OCS fluxes shows that some combination of a larger tropical oceanic source and vegetative sink resolves the budget". Kooijmans et al. (2021) also show that considering a variable atmospheric OCS concentration reduces the vegetation sink in the tropics, meaning that a smaller tropical OCS source would be needed to close the budget. – *See above.*

L392: "such as calculating OCS uptake using a constant LRU value of 1.6 is not representative of reality". It could be interesting to give the range of values proposed for LRU in the literature, to illustrate how LRU values can vary. – *Updated. Line 565-575.*

L394: "Our estimation of vegetative uptake in this work does not replicate OCS uptake universally and it is unclear if this is due to localised differences in LRU or on the GPP fields themselves". Could you provide a global map of vegetation OCS uptake obtained with your approach using a constant LRU value and compare it to similar maps found in the literature (Berry et al., 2013; Kooijmans et al., 2021; Maignan et al., 2021) to analyse the spatial distribution of the fluxes? Could you also provide a map of TOMCAT simulated atmospheric CO2 concentrations used to compute vegetation OCS uptake? – *See the supplement for these additions.*

L396: "the distribution is based on work by Kettle et al. (2002) and has since been updated, for example by Ogée et al. (2016)". It has also been updated by Sun et al. (2015). It could be interesting to compare the spatial distribution of your soil OCS fluxes to other maps based on the mechanistic approach from Ogée et al. (2016) (Kooijmans et al., 2021; Abadie et al., 2022). – *See the supplement for these additions.*

6. Conclusion:

L453: "Therefore, further study following on from this work will be to derive an a posteriori set of fluxes using an inversion scheme based on an up-to-date prior, and surface observations and a dataset containing vertical information near the surface". Important drawbacks are acknowledged by the authors as this work does not rely on an optimization framework and uses out-of-date OCS fluxes. These drawbacks should explicitly appear in the abstract. – *Abstract has been mostly rewritten.*

**Minor comments:**

L19: "To compensate for this larger vegetative sink". I would not use "larger" here as it has not been explained yet that it is larger than the vegetation OCS uptake from Kettle et al. (2022). – *Updated.*

L30: Is this really "Hawaiin" and not "Hawaian" (replace everywhere if needed)? – *Updated to Hawaiian everywhere.*

L56: Please replace "Vegetative uptake is the most important atmospheric sink of OCS" by "Vegetative uptake is the most important sink of atmospheric OCS". – *Updated. Line 74.*

L107: Please develop the abbreviation HITRAN. – *Updated. Line 144-145.*

Table 1: Please replace "Barrow" by "Utqiagvik (formerly Barrow)" and "Cape Grim" by "Kennaook / Cape Grim". For PSA and SPO stations, please precise "Antarctica (United States)". – *Updated.*

L135: Should "surface emission fields" be replaced by "surface flux fields" as surface OCS fluxes are not only sources? – *Updated. Line 175.*

L136: Please replace "six sources and three sinks" by "six net sources and three net sinks" as soils can be both a sink or a source of OCS for example. Please also name the sinks and sources here. – *Updated*

L147: Remove "an" in this sentence "Eleven anthropogenic sources of OCS were an quantified by Zumkehr et al. (2018)". – *Updated*

L175: Please develop the abbreviation WATCH. – *Updated*

L235: "so we only compare the main simulations". Please precise that it is TOMCATocs. – *Updated*

L301: Please remove "Gg" in the following "from 17.7 Gg ppt to 3.4 ppt". – *Updated*

Figure1: Please improve the resolution of the figure to be able to read the RMSE scores. – *Updated*

**RC2 (from attached pdf document)**

**Section 1**
L19: confusing, since you say "towards the lower end", so expect "smaller". – *Abstract has been mostly rewritten.*

L39: define lifetime: global burden/stratospheric loss or stratospheric burden/loss? – Updated. Line 48-50.

L55: confusing. "DMS accounts for OCS oceanic emissions". DMS is emitted and an uncertain fraction is oxidized to OCS. refer recent findings: Jernigan, C. M., Fite, C. H., Vereecken, L., Berkelhammer, M. B., Rollins, A. W., Rickly, P. S., Novelli, A., Taraborrelli, D., Holmes, C. D., & Bertram, T. H. (2022). Efficient production of carbonyl sulfide in the low-NO x oxidation of dimethyl sulfide . Geophysical Research Letters, x, 1–11. https://doi.org/10.1029/2021gl096838 - *Updated. Line 71-73.*

L66: recent update: @article{abadie2022global, title={Global modelling of soil carbonyl sulfide exchanges}, Kooijmans, L. M. J., Cho, A., Ma, J., Kaushik, A., Haynes, K. D., Baker, I., Luijkx, I. T., Groenink, M., Peters, W., Miller, J. B., Berry, J. A., Ogée, J., Meredith, L. K., Sun, W., Kohonen, K. M., Vesala, T., Mammarella, I., Chen, H., Spielmann, F. M., … Krol, M. (2021). Evaluation of carbonyl sulfide biosphere exchange in the Simple Biosphere Model (SiB4). Biogeosciences, 18(24), 6547–6565. https://doi.org/10.5194/bg-18-6547-2021- *Updated. Line 85-88.*

L84: I find this sentence strange. Normally one would use OCS vegetative uptake to estimate GPP.....so this becomes rather confusing. – *Removed this line. While we do clarify that the LRU approach does suitably estimate vegetative uptake, this had been shown previously.*

L92: are – *Updated.*

**Section 2**
**Section 3**
L153: do I miss the chemistry terms here (OH, and photolysis)? – *Line 166-170.*

L169: I think you have to be clear about units here. GPP and Fcos differ by orders of magnitude... – *Updated to be more clear on units. All appropriate scaling is performed in the calculation. Line 223-227.*

L178: I think Ma et al. showed that is even more important to account for OCS mixing ratios. Now it remains unclear how [OCS] is used to convert GPP to F_OCS – *We use OCS from the model at each timestep. The text has been updated to make this more clear. For example on Line 228.*

L194: fluxes?

L195: uptake?

L197: I understand, but should be something like: due to the resulting improvements ....

L206: ocean emissions

L219: I think OH loss is mostly tropospheric. – updated.

Table 2: I read the Ma et al. table well, this is the imbalance in their budget.... – *This has been updated in Table 2.*

L236: did not read that in section 2.1, nor in section 3.1 – updated.

**Section 4**

L237: This does not make sense. Why not compare both simulations? – As the budgets of the other model runs are negative, it would make correction throughout the entire atmosphere very challenging.

L243: mm, this was not stated in the method section. Actually, this belongs in the method section, and the 10 year initialization is mentioned in the TOMCAT_CON description. – Updated.

L248: please do not repeat the method section here. – Updated.

L250: What I no not understand why monthly means are compared, while ACE-FTS is co-sampled. I think co-sampling is important at the surface. Also, I find the metric max-min to estimate the monthly error in observations misleading. – Figure 1 updated to use standard deviation.

L272: This improvement is rather disappointing. This implies that budget terms and their seasonality are not OK yet? – *It is not an excellent improvement. However, when compared also to TOMCATSOTA, we see that it is a fairly good improvement. While TOMCAT$_{SOTA}$ does well in the NH in terms of SCA, it performs poorly at LEF and HFM.*

L279: Here it is really vital to investigate the modeled diurnal cycle. And during strong uptake OCS goes down, and the LRU formulation might break down. This is very handwaving argumentation. – *This sentence was removed. But our work does indeed suggest the LRU approach still underestimates at heavily vegetated regions.*

L282: Well, also handwaving: was JULES NEE validated with flux measurements at this location (of at Harvard Forest)? – *JULES GPP was validated against FLUXNET. The text has been updated accordingly. Line 405-407.*

L287: did you check this? Or is this speculation. Coudl anthropogenic emissions influence MHD is some month?

L289: The big question hanging here is: is this due to model shortcomings, or due to the applied COS fluxes. Given the fact that inverse models capture the seasonal cycle at MHD, you would be tempted to say that fluxes are not yet optimal, e.g. too high ocean emissions over the Atlantic... – *This is likely the cause and text has been updated. Line 415-419.*

L298: did you check (e.g. with tagged tracer run), or can you provide a reference. – *Updated as above.*

L301: ??

Figure 1: increase font size... AND What are the Blue error bars? CAPTION: is. – Figure 1 has been updated and the blue bars now represent standard deviation.

L337: Again: here you speculate. It would be good to perform tagged tracer runs to backup these statements. – This was not considered in the scope of corrections.

L342: I assume you mean something like the maximum concentration in the vertical.

L343: ?? If ACE connot measure, this does not make sense. – *Removed.*

L347: There is nothing between these two seaons? – *Updated Line 473.*

L359: There is abundant speculation in the paper. Some things really need to be checked better ....One thing is this "faster removal". What about reducing the photolysis rates to see whether the disagreement disappears? – *The author improves on argumentation and discussion. We include an additional model run to test the comments on photolysis loss in the stratosphere.*

L364: Not sure if this is the ACE uncertainty, since this is the variability in observations. Does ACE provide an observational error? – *Amended to be standard deviation.*

L366: see comment above

**Section 5**

L386: I think you overstate the quality of your simulations here. There are significant remaining deviations that apparently are resolved when emissions are optimized. Now the seasonal cycles are sifted and mostly underestimated (Figure 1). – *Amended the text to suggest that our flux also points towards a missing tropical source, like the inversion studies referenced. Less so that our model compares as well as theirs.*

L388: ???? – Removed.

L394: and sensitive to the unknown OCS mixing ratio at the uptake locations... – *Updated. Line 565-566.*

L397: see also new paper Abadie (2022) – *Updated. Line 570-571.*

L410: Again,, OH mostly acts in the troposphere. – *Updated this sentence. Line 539-541.*

L413: Again, a speculation that can be easily tested.. – *Removed this sentence.*

**Section 6**

L448: d – *Updated. Line 613.*

---

## Author Response (AR1)

Firstly, thank you to the anonymous referees for taking the time to evaluate this work.

Here we address all major or general comments made by the two referees.

Please find all responses to minor comments specific to the text in the supplementary document attached.

RC1

1. The main concern from RC1 here is the insufficient use of recent references in the introductory material, as well as using primarily out-of-date fluxes and data in the model.

"References to recent studies are missing in the introduction." - We address these concerns by firstly updating the introduction to include a broader review of available literature to better represent recent scientific findings for OCS.

"Using the inventory of Kettle et al. (2002) is also a major weakness as many studies have provided new OCS flux estimates since. Therefore, the conclusion that TOMCAT$_{ocs}$ gives better results than TOMCAT$_{con}$ does not seem relevant when TOMCAT$_{con}$ is based on out-of-date estimates (except for the anthropogenic emissions from Zumkehr et al., 2018). " - To address this concern (and others detailed in 1.) we include an additional model run for comparison with TOMCAT$_{OCS}$. This new model is called TOMCAT$_{SOTA}$ and makes use of flux inventories developed between 2017 - 2022, including work by the following: Stinecipher et al. (2019), Zumkehr et al. (2018), Lennartz et al. (2017, 2020) and Maignan et al. (2021). See the text for more specific detail. This new model run provides an additional comparison for TOMCAT$_{OCS}$, one that is more relevant to recent findings and strengthens the conclusions made about the quality of TOMCAT$_{OCS}$.

"In Section 3.3, why choosing to use a single constant LRU value while several studies provide PFT-dependent LRU values?" - On a global scale it seems unlikely that a varying LRU would make a large difference, which is highlighted by Hilton et al. (2017): "the methodological simplification of treating LRU as a constant introduces considerably less error than GPP uncertainty (as estimated by inter-model GPP differences).". While this will be included in the scope of future work, the primary goal was to present a suitable model that compares well with measurements.

" In Section 3.4, the OCS fluxes that were adjusted to obtain a balanced budget should be contrasted with more recent estimates. For example, choosing to scale CS2 oceanic emissions to 439 GgS/y is not supported by Lennartz et al. (2020) who estimated a total source of 70 GgS/y from CS2. An oxic soil budget of 322 GgS/y is also not in line with the recent estimates of Kooijmans et al. (2021) and Abadie et al. (2022) based on the mechanistic soil model of Ogée et al. (2016). " – Firstly, thank you for bringing these works to my attention, the method and discussion sections feature more references to recent literature in line with this request. However, from an actionable perspective this comment poses several challenges. Firstly, the use of CS2 is utilised purely due to its spatial distribution over the tropics from the Kettle et al. (2002) inventory – this is mentioned several times in the text. Secondly, while Kooijmans et al. (2021) does suggest soil uptake of 89 – 146 Gg S/y using SIB4, this paper was published in December 2021 during the final stages of preparing this manuscript. Furthermore, the discussion for Abadie et al. (2022) stared in November 2021, again during final preparations for this work and was actually published nearly 2 months following (11 May 2022) the initial

submission of this work (18 March 2022). Finally, it is worth highlighting that the combined biosphere flux presented in this work, used in TOMCAT$_{OCS}$, is inline with inversion studies by Ma et al. (2021) and Remaud et al. (2022).

"More recent studies should be added in Table 2 to compare to the OCS budget from this work, such as Maignan et al. (2021), Kooijmans et al. (2021), Remaud et al. (2022)." – Remaud et al. (2022) is now also included.

2. The main concern from RC1 here is that optimisation of the fluxes would be better achieved using an inversion framework and that balancing the budget is unnecessary.

"The scaling of OCS fluxes to better match estimates made after Kettle et al. (2002) and to obtain a balanced OCS budget seems quite arbitrary. Such adjustments should be made using an inversion framework as done in Ma et al. (2021) or in Remaud et al. (2022). Without an analytical inverse system that optimizes the fluxes, why aiming at a balanced COS budget? A balanced OCS budget is also not required if analyzing the detrended OCS concentrations." – This is a fair statement but does not factor in the accessibility to an inverse framework at the time of undertaking the research. Also, while a balanced budget is indeed under debate (less so in the past 5 or so years), it has been shown for the majority of the period between 2004 and 2018, that the OCS budget is indeed in balance or with a weak trend, depending on latitude.

"Moreover, this scaling assumes that the OCS flux spatial distribution of each component is not modified compared to the control inventory, which might not agree with flux distribution obtained in more recent studies." – This is indeed a limitation of the method of scaling in this work and is acknowledged by the author(s). The compromise proposed here is to have included the third model run, as discussed in 1. This provides an additional comparison with up-to-date spatial distribution of fluxes, including maps presented in the supplementary material. The quality of the vertical comparison with ACE-FTS suggests the troposphere, surface and atmospheric fluxes are modelled well.

3. The main concerns here were regarding the calculation of vegetative uptake of OCS, particularly CO2 and OCS concentration.

"Not considering interannual variations is a strong assumption that should at least be better justified. The study from Chen et al. (2017) does not conclude that interannual variability in GPP amplitude can be neglected. GPP interannual variability could easily be included in this work as GPP is modelled by JULES. Considering only the year 2010 does not reflect the yearly increase in atmospheric CO2 concentration and the fertilization effect." It was assumed that including variability in CO2 concentration is likely to be a smaller source of error or variation in F$_{OCS}$ than other factors, such as GPP product and OCS concentration. Like LRU this would be included in the future scope of this work.

"Otherwise, could the impact of not considering OCS flux interannual variability be quantified? For example, OCS vegetation uptake could be defined as a first order relationship with OCS mixing ratio. Therefore, inter-annual variations in OCS vegetation flux might have a strong impact on the simulated atmospheric OCS concentrations." – The calculation of vegetative uptake is actually calculated at each time-step in the model (every 6 hours). The text has been updated to make this clearer. But we do establish a first order relationship between vegetative uptake and OCS concentration.

4. RC1 main points here are that the paper lacked direction and conclusion, and that some of the comparing TOMCAT$_{OCS}$ to TOMCAT$_{CON}$ was a challenge and a bit arbitrary.

"It is not clear what the goal of this study is, and the title is confusing." – The goal of the study was to build a model that simulates OCS well compared to novel satellite observations and surface measurements. In turn to extract information about the implications of the OCS budget estimated here and how this compares with recent literature. TOMCAT$_{SOTA}$ helps provide a benchmark for recent work and how a collection of bottom-up estimates does not necessarily yield sensible seasonality at the surface. Hence, highlighting there is still substantial uncertainties in the overall OCS budget. This work agrees with the missing source originating from the tropics but falls short of concluding its exact origin. As mentioned in the text, future work will aim to use an inverse system in combination with data that offers vertical information, such as ACE-FTS, but preferably also in the lower troposphere, like NDACC. Connecting the troposphere to the surface could be the key to improving surface flux estimates.

We have recommended a change in title from:

Modelling atmospheric carbonyl sulfide using gross primary productivity to constrain vegetative uptake

to

Constraining the budget of atmospheric carbonyl sulfide using a 3-D chemical transport model

"Should this work focus more on the advantage of using ACE-FTS compared to other available OCS concentration observations? Or on the information that can be retrieved from ACE-FTS about the modelling of OCS atmospheric sinks?" – The quality of TOMCAT$_{OCS}$ comparison with ACE-FTS is highlighted more so in the revised manuscript, with an underlying message that a good comparison throughout most of the atmosphere suggests that the gradients of OCS are well represented by the model. Therefore, so are the surface and atmospheric fluxes driving it.

RC2

"First, when the results are described in the paper, often hand-waving argumentation is used to explain the deviations between model and observations. "Likely caused", "could be attributed". Here, we have to believe the judgement of the authors, since rarely additional arguments are presented." – The author has improved on this by either removing speculation or better comparing and referencing conclusions, particularly in the results and discussion sections.

"Likewise, the underestimation of modelled OCS in the tropical stratosphere is explained by too fast removal. These observations call for additional simulations to verify whether speculations hold true. Two suggestions here: (1) a simulation with tagged tracers to be used in a more detailed analysis of e.g. seasonal cycles (2) a simulation with reduced photochemical removal in the tropical stratosphere. Presentation of the results would give the paper more body." – We include an additional model simulation for 2010 only that reduces atmospheric photolysis. The intention of this simulation is to test if this does resolve the issues in the stratosphere. What we find is while it does reduce negative bias in the Southern Hemisphere, it introduces model bias elsewhere, i.e. positive in the Northern Hemisphere tropics. So we draw the conclusion that the convection scheme or transport is causing the issue. An experiment

with 0.75 photolysis rate is presented in the manuscript and an additional one with 0.5 photolysis is shown in the supplement.

"It remains unclear what is taken for the OCS mixing ratios here." – As described in point 3 of RC1, we utilise the OCS concentration at each time-step in the model to calculate the vegetative flux. The model is initialised at the start of the spin-up period (1994) using 500 ppt however.

"The paper therefore misses quite some recent references that are relevant for the work. The authors should update the reference list (and discussions) with more recent papers." – The introductory material has been updated to include more recent literature. More frequent use of up-to-date references are used throughout the text.

**Minor Comments**

**RC1**

Abstract:

L25: "At the surface, the model captures background concentrations at most of the surface sites to within the maximum and minimum of the seasonal measurements". It does not seem to be a strong condition to satisfy. It might be better to highlight results on the seasonal cycle amplitude or the phase. – *The error bars represented in Figure 1 are now standard deviation, rather than maximum and minimum. Results of SCA are highlighted in the new abstract.*

1.Introduction:

L49: "The main source of atmospheric OCS is oceanic emission". This could be replaced by "one of the main sources" as anthropogenic emissions are also a major source of OCS. For example, Zumkher et al. (2018) estimated an anthropogenic OCS source of about 400 GgS/y. Aydin et al. (2020) suggested that this estimate could be underestimated with an anthropogenic OCS source of about 600 GgS/y. – *Updated. Line 64.*

L57: "with estimates ranging from 210 to 2400 Gg S yr-1 (Kettle et al., 2002; Sandoval-Soto and Stanimirov, 2005; Suntharalingam et al., 2008; Berry et al., 2013; Glatthor et al., 2015; Kuai et al., 2015; Launois et al., 2015b; Ma et al., 2021)". Add references to more recent studies such as Maignan et al. (2021), Kooijmans et al. (2021), Remaud et al. (2022) for vegetation OCS uptake estimates. – *Updated. Line 75-77.*

L61: "OCS hydrolysis also occurs in soil, again catalysed by carbonic anhydrase". Note that OCS can also be consumed by other enzymes in soils, such as nitrogenase, CO dehydrogenase, or CS2 hydrolase (Smith and Ferry, 2000; Masaki et al., 2021). – *Updated. Line 81-82.*

L63: "with an estimated annual loss of 127-355 Gg S (Kettle et al., 2002; Montzka et al., 2007; Berry et al., 2013; Glatthor et al., 2015; Kuai et al., 2015)". More recent studies should be mentioned, such as Kooijmans et al. (2021) and Abadie et al. (2022) that lead to smaller soil OCS budgets. – *Updated. Line 83.*

L65: "Soil has also been observed to act as an emitter of OCS in warm conditions (Maseyk et al., 2014)". This was not observed only for warm conditions. OCS emissions have also been related to soil types (Whelan et al., 2013), nitrogen content, light radiations reaching the soil surface (Spielmann et al., 2019; Kitz et al., 2020). – *Updated. Line 85-88.*

L69: "the latter of which has been used as a benchmark for more recent studies". Please add the references of the studies. – *Updated. Line 92-93.*

2. Observations:

Section 2.1: Please provide and detail the uncertainties associated with ACE-FTS retrievals. – *Updated. Line 145-147.*

3. Chemical transport modelling of OCS:

Section 3.1: What is the timestep used to run the TOMCAT model? – *6 hours. Updated. Line 172-173.*

L139 to L224: Please make it clearer which fluxes have been used for TOMCATcon and which one have been used for TOMCATocs. – *Updated. TOMCAT$_{CON}$ uses inventory in Section 3.2 and TOMCAT$_{OCS}$ uses inventory in Section 3.3.*

L153: "The three sink terms are an oceanic sink, soil uptake and a vegetative sink". OCS photolysis in the stratosphere and OCS oxidation by OH radical in the troposphere should also be included in OCS sinks, as atmospheric OCS reactions are not explained before in section 3.1. – *We include a summary of OH and Photolysis loss in Section 3.1, before describing all the fluxes. It is clear the same scheme is used for all model runs. Line 166-170.*

Equation 1: Precise the unit for each term of this equation. What is used for OCS background concentration? – *Updated. Line 222-226.*

L173: Please replace "LRU is the normalised ratio of OCS assimilation rates to CO2 at the leaf-scale. This is then normalized by background concentrations of the two gases" by "LRU is the ratio of OCS assimilation rates to CO2 at the leaf-scale, both normalized by their respective concentration". – *Updated. Line 223-224.*

L180: "but is slightly under half that of the largest estimation of 1115 Gg S in Table 2 from Montzka et al. (2007)". Launois et al. (2015) estimated a larger plant OCS uptake than Montzka et al. (2007) for the ORCHIDEE land surface model. – *Updated. Line 242.*

L204: "at Northern Hemisphere (NH) NOAA-ESRL sites". Please precise which sites and whether they receive air masses mainly coming from the ocean. – *Updated. Line 277-278.*

Table 2: Why were more recent studies not included in this table for comparison? Such as Maignan et al. (2021), Kooijmans et al. (2021), Remaud et al. (2022). – *Updated to include Remaud et al. (2022).*

4. Results:

L243 to L245: "TOMCATCON was initialised using OCS values in each grid box from TOMCATOCS, after 10 years (1994 – 2003) spin-up. Only 2004 monthly mean mixing ratios from TOMCATCON have been included, as this flux inventory has a 245 net negative budget and therefore a negative trend over longer periods". Should this be in Section 3 as it is related to the method? – *Moved to section 3. Line 187.*

L244: "Only 2004 monthly mean mixing ratios from TOMCATCON have been included, as this flux inventory has a net negative budget and therefore a negative trend over longer periods". Please precise the net negative budget. The trend in atmospheric COS concentrations should not be an issue if you remove the trend and compare the detrended atmospheric concentrations. – *Updated. Line 188. Figure 2 now presents monthly anomalies.*

L249: "Error bars associated with the observations represent the maximum and minimum values for each month at every site". Representing the standard deviation would be a better indication of the uncertainty of the mean value. – *Updated. Figure 1 error bars are now represented by standard deviation.*

L253: "Comparisons between TOMCATocs and TOMCATcon are shown here to emphasise the improvements made by the flux inventory developed in this study". How could the improvements obtained with TOMCATocs on atmospheric OCS concentrations be compared to the improvements made when using inversion systems such as in Ma et al. (2021) and Remaud et al. (2022)? – *We compare TOMCATOCS to a new model run TOMCATSOTA, which utilises fluxes that are more up-to-date in the literature and similar or the same as those used in as prior fluxes by Ma et al. (2021) and Remaud et al. (2022). See discussion and supplement.*

L254: "The root mean square error (RMSE) for the entire period is shown for each site, alongside the seasonal cycle amplitude (SCA)". Precise that in the following you also compare the phases of observed and simulated seasonal cycles. – *SCA is now displayed in Figure 2.*

L255: "Generally, there is an improvement in RMSE across all the sites, but in some cases, there is a degradation, which is mostly attributed to background concentration, rather than the model's ability to capture a suitable seasonal cycle, hence both are shown." By "background concentration", do you mean the average concentration? If so, could you please show that the degradation in RMSE is due to the average concentration? – *I did mean average concentration. Figure 1 shows $TOMCAT_{OCS}$ and $TOMCAT_{CON}$ monthly mean model values, but Figure 2 shows monthly anomalies. So we are able to distinguish the impact from poor estimation of average concentration and of the seasonality.*

L264: "This seasonal cycle resembles that of CO2, hence GPP is a suitable proxy for calculating OCS uptake". Please rephrase as similar seasonal cycle is not reason enough to use GPP as a proxy of vegetation OCS uptake. – *I agree. This was rephrased. Line 377.*

L275: "Here we show realistic amplitudes in the seasonal cycle from TOMCATOCS, 76 ppt at LEF and 71 ppt at HFM, compared to observed values of 123 ppt and 128 ppt, respectively". Please rephrase as the SCA at these two sites are still largely underestimated. – *Rephrased. Line 399-401.*

L279 to 282: The constant LRU value used in this study could be compared to other LRU estimates for the same vegetation types found at LEF and HFM to see if it could be underestimated. If a constant OCS mixing ratio was used to compute OCS vegetation uptake, this could also affect the SCA. – *I have updated the text in places to make it clear we are using OCS concentration from online within the model. A brief discussion of LRU in the literature is made in Section 5: Line 565-570.*

L329: "OCS values decline above and below the UTLS due to removal by photosynthesis at the surface". Soils can also absorb OCS at the surface. – *Updated. Line 456-457.*

L345: "potentially attributed to slower surface OCS uptake". It could also be due to an underestimated surface OCS uptake. – *Updated. Line 472-473.*

Figure 2: What could explain the net distinction between higher mixing ratios in NH compared to SH found in TOMCAT in JJA and SON? – *Weaker removal of OCS in the SH.*

L364: "this suggests the upper atmospheric sinks are modelled well by TOMCATocs". Isn't it in contradiction with the steeper gradient of TOMCATocs in the stratosphere mentioned above? – *Slightly rephrased to play down the performance. Line 492-493.*

Section 4.2 attributes model-observation mismatches to OCS sources or sinks, what about the potential mismatches from TOMCAT transport? – *This is mentioned in Section 4.3. Line 516.*

5. Discussion:

L376: "the TOMCATocs simulations of atmospheric OCS concentrations and the vegetative flux, which are dependent on one another in the model". The simulated OCS concentrations are dependent on vegetation OCS fluxes that were transported, but it is not said which OCS concentrations are used to compute the vegetation OCS fluxes. – *OCS concentration from the model is used in the calculation of vegetative uptake. See Line 228.*

L385: "inverse modelling of OCS fluxes shows that some combination of a larger tropical oceanic source and vegetative sink resolves the budget". Kooijmans et al. (2021) also show that considering a variable atmospheric OCS concentration reduces the vegetation sink in the tropics, meaning that a smaller tropical OCS source would be needed to close the budget. – *See above.*

L392: "such as calculating OCS uptake using a constant LRU value of 1.6 is not representative of reality". It could be interesting to give the range of values proposed for LRU in the literature, to illustrate how LRU values can vary. – *Updated. Line 565-575.*

L394: "Our estimation of vegetative uptake in this work does not replicate OCS uptake universally and it is unclear if this is due to localised differences in LRU or on the GPP fields themselves". Could you provide a global map of vegetation OCS uptake obtained with your approach using a constant LRU value and compare it to similar maps found in the literature (Berry et al., 2013; Kooijmans et al., 2021; Maignan et al., 2021) to analyse the spatial distribution of the fluxes? Could you also provide a map of TOMCAT simulated atmospheric CO2 concentrations used to compute vegetation OCS uptake? – *See the supplement for these additions.*

L396: "the distribution is based on work by Kettle et al. (2002) and has since been updated, for example by Ogée et al. (2016)". It has also been updated by Sun et al. (2015). It could be interesting to compare the spatial distribution of your soil OCS fluxes to other maps based on the mechanistic approach from Ogée et al. (2016) (Kooijmans et al., 2021; Abadie et al., 2022). – *See the supplement for these additions.*

6. Conclusion:

L453: "Therefore, further study following on from this work will be to derive an a posteriori set of fluxes using an inversion scheme based on an up-to-date prior, and surface observations and a dataset containing vertical information near the surface". Important drawbacks are acknowledged by the authors as this work does not rely on an optimization framework and uses out-of-date OCS fluxes. These drawbacks should explicitly appear in the abstract. – *Abstract has been mostly rewritten.*

**Minor comments:**

L19: "To compensate for this larger vegetative sink". I would not use "larger" here as it has not been explained yet that it is larger than the vegetation OCS uptake from Kettle et al. (2022). – *Updated.*

L30: Is this really "Hawaiin" and not "Hawaian" (replace everywhere if needed)? – *Updated to Hawaiian everywhere.*

L56: Please replace "Vegetative uptake is the most important atmospheric sink of OCS" by "Vegetative uptake is the most important sink of atmospheric OCS". – *Updated. Line 74.*

L107: Please develop the abbreviation HITRAN. – *Updated. Line 144-145.*

Table 1: Please replace "Barrow" by "Utqiagvik (formerly Barrow)" and "Cape Grim" by "Kennaook / Cape Grim". For PSA and SPO stations, please precise "Antarctica (United States)". – *Updated.*

L135: Should "surface emission fields" be replaced by "surface flux fields" as surface OCS fluxes are not only sources? – *Updated. Line 175.*

L136: Please replace "six sources and three sinks" by "six net sources and three net sinks" as soils can be both a sink or a source of OCS for example. Please also name the sinks and sources here. – *Updated*

L147: Remove "an" in this sentence "Eleven anthropogenic sources of OCS were an quantified by Zumkehr et al. (2018)". – *Updated*

L175: Please develop the abbreviation WATCH. – *Updated*

L235: "so we only compare the main simulations". Please precise that it is TOMCATocs. – *Updated*

L301: Please remove "Gg" in the following "from 17.7 Gg ppt to 3.4 ppt". – *Updated*

Figure1: Please improve the resolution of the figure to be able to read the RMSE scores. – *Updated*

**RC2 (from attached pdf document)**

**Section 1**
L19: confusing, since you say "towards the lower end", so expect "smaller". – *Abstract has been mostly rewritten.*

L39: define lifetime: global burden/stratospheric loss or stratospheric burden/loss? – Updated. Line 48-50.

L55: confusing. "DMS accounts for OCS oceanic emissions". DMS is emitted and an uncertain fraction is oxidized to OCS. refer recent findings: Jernigan, C. M., Fite, C. H., Vereecken, L., Berkelhammer, M. B., Rollins, A. W., Rickly, P. S., Novelli, A., Taraborrelli, D., Holmes, C. D., & Bertram, T. H. (2022). Efficient production of carbonyl sulfide in the low-NO x oxidation of dimethyl sulfide . Geophysical Research Letters, x, 1–11. https://doi.org/10.1029/2021gl096838 - *Updated. Line 71-73.*

L66: recent update: @article{abadie2022global, title={Global modelling of soil carbonyl sulfide exchanges}, Kooijmans, L. M. J., Cho, A., Ma, J., Kaushik, A., Haynes, K. D., Baker, I., Luijkx, I. T., Groenink, M., Peters, W., Miller, J. B., Berry, J. A., Ogée, J., Meredith, L. K., Sun, W., Kohonen, K. M., Vesala, T., Mammarella, I., Chen, H., Spielmann, F. M., … Krol, M. (2021). Evaluation of carbonyl sulfide biosphere exchange in the Simple Biosphere Model (SiB4). Biogeosciences, 18(24), 6547–6565. https://doi.org/10.5194/bg-18-6547-2021- *Updated. Line 85-88.*

L84: I find this sentence strange. Normally one would use OCS vegetative uptake to estimate GPP.....so this becomes rather confusing. – *Removed this line. While we do clarify that the LRU approach does suitably estimate vegetative uptake, this had been shown previously.*

L92: are – *Updated.*

**Section 2**
**Section 3**
L153: do I miss the chemistry terms here (OH, and photolysis)? – *Line 166-170.*

L169: I think you have to be clear about units here. GPP and Fcos differ by orders of magnitude... – *Updated to be more clear on units. All appropriate scaling is performed in the calculation. Line 223-227.*

L178: I think Ma et al. showed that is even more important to account for OCS mixing ratios. Now it remains unclear how [OCS] is used to convert GPP to F_OCS – *We use OCS from the model at each timestep. The text has been updated to make this more clear. For example on Line 228.*

L194: fluxes?

L195: uptake?

L197: I understand, but should be something like: due to the resulting improvements ....

L206: ocean emissions

L219: I think OH loss is mostly tropospheric. – updated.

Table 2: I read the Ma et al. table well, this is the imbalance in their budget.... – *This has been updated in Table 2.*

L236: did not read that in section 2.1, nor in section 3.1 – updated.

**Section 4**

L237: This does not make sense. Why not compare both simulations? – As the budgets of the other model runs are negative, it would make correction throughout the entire atmosphere very challenging.

L243: mm, this was not stated in the method section. Actually, this belongs in the method section, and the 10 year initialization is mentioned in the TOMCAT_CON description. – Updated.

L248: please do not repeat the method section here. – Updated.

L250: What I no not understand why monthly means are compared, while ACE-FTS is co-sampled. I think co-sampling is important at the surface. Also, I find the metric max-min to estimate the monthly error in observations misleading. – Figure 1 updated to use standard deviation.

L272: This improvement is rather disappointing. This implies that budget terms and their seasonality are not OK yet? – *It is not an excellent improvement. However, when compared also to TOMCATSOTA, we see that it is a fairly good improvement. While TOMCAT$_{SOTA}$ does well in the NH in terms of SCA, it performs poorly at LEF and HFM.*

L279: Here it is really vital to investigate the modeled diurnal cycle. And during strong uptake OCS goes down, and the LRU formulation might break down. This is very handwaving argumentation. – *This sentence was removed. But our work does indeed suggest the LRU approach still underestimates at heavily vegetated regions.*

L282: Well, also handwaving: was JULES NEE validated with flux measurements at this location (of at Harvard Forest)? – *JULES GPP was validated against FLUXNET. The text has been updated accordingly. Line 405-407.*

L287: did you check this? Or is this speculation. Coudl anthropogenic emissions influence MHD is some month?

L289: The big question hanging here is: is this due to model shortcomings, or due to the applied COS fluxes. Given the fact that inverse models capture the seasonal cycle at MHD, you would be tempted to say that fluxes are not yet optimal, e.g. too high ocean emissions over the Atlantic... – *This is likely the cause and text has been updated. Line 415-419.*

L298: did you check (e.g. with tagged tracer run), or can you provide a reference. – *Updated as above.*

L301: ??

Figure 1: increase font size... AND What are the Blue error bars? CAPTION: is. – Figure 1 has been updated and the blue bars now represent standard deviation.

L337: Again: here you speculate. It would be good to perform tagged tracer runs to backup these statements. – This was not considered in the scope of corrections.

L342: I assume you mean something like the maximum concentration in the vertical.

L343: ?? If ACE connot measure, this does not make sense. – *Removed.*

L347: There is nothing between these two seaons? – *Updated Line 473.*

L359: There is abundant speculation in the paper. Some things really need to be checked better ....One thing is this "faster removal". What about reducing the photolysis rates to see whether the disagreement disappears? – *The author improves on argumentation and discussion. We include an additional model run to test the comments on photolysis loss in the stratosphere.*

L364: Not sure if this is the ACE uncertainty, since this is the variability in observations. Does ACE provide an observational error? – *Amended to be standard deviation.*

L366: see comment above

**Section 5**
L386: I think you overstate the quality of your simulations here. There are significant remaining deviations that apparently are resolved when emissions are optimized. Now the seasonal cycles are sifted and mostly underestimated (Figure 1). – *Amended the text to suggest that our flux also points towards a missing tropical source, like the inversion studies referenced. Less so that our model compares as well as theirs.*

L388: ???? – Removed.

L394: and sensitive to the unknown OCS mixing ratio at the uptake locations... – *Updated. Line 565-566.*

L397: see also new paper Abadie (2022) – *Updated. Line 570-571.*

L410: Again,, OH mostly acts in the troposphere. – *Updated this sentence. Line 539-541.*

L413: Again, a speculation that can be easily tested.. – *Removed this sentence.*

**Section 6**
L448: d – *Updated. Line 613.*

---

## Referee Report (RR1)

[referee-annotated manuscript omitted]

---

## Author Response (AR2)

Constraining the budget of atmospheric carbonyl sulfide using a 3-D chemical transport model

**Response to Reviewers' Comments**

We thank the reviewers for taking the time to evaluate our manuscript and for their positive and helpful comments. These comments are reproduced below in *italics*, followed by '>>' and our responses.

References to line numbers for new specific sentences based on Reviewer comments refer to the marked-up version of the new manuscript.

**Reviewer 1**

*The authors have made a large effort to answer a majority of the comments; I notably appreciate the addition of Figure 2 representing seasonal anomalies of OCS concentrations at NOAA sites, as well as the various OCS flux maps provided in the Supplementary Material. However, although the manuscript has been improved and is now clearer and easier to follow, not all answers were satisfying and there are still several points that need to be correctly addressed.*

**General comments**

*State-of-the-art fluxes*

*It's a good start to consider state-of-the-art (SOTA) fluxes, but comparing the concentrations derived from the transport of the prior SOTA fluxes to the concentrations derived from the transport of the optimized TOMCAT-OCS fluxes does not make sense. You have to think about the scientific message that you want to convey, which is probably not that you can better match observed concentrations by manually scaling outdated fluxes.*

*My initial comment was the following one: "How could the improvements obtained with TOMCATocs on atmospheric OCS concentrations be compared to the improvements made when using inversion systems such as in Ma et al. (2021) and Remaud et al. (2022)?"*

*Indeed, comparing with concentrations computed based on the optimized fluxes from Ma et al. (2021) and/or Remaud et al. (2022) would have been more fruitful.*

>> "*You have to think about the scientific message that you want to convey, which is probably not that you can better match observed concentrations by manually scaling outdated fluxes.*" – We have considered this point and have included some additional comments in the discussion and conclusion on what the difference between TOMCAT$_{OCS}$ and TOMCAT$_{SOTA}$ means in terms of the fluxes. For example, why is the mechanistic model (the SOTA vegetative flux) not capturing the seasonality adequately? What underlying difference between the two methods is causing a dissimilar seasonality at surface sites?

"*Indeed, comparing with concentrations computed based on the optimized fluxes from Ma et al. (2021) and/or Remaud et al. (2022) would have been more fruitful.*" – It is worth noting that while these are optimised fluxes, they do not provide information on bottom-up processes. Hence, we direct the reader to consider the above, i.e., the difference between the LRU method and the mechanistic approach.

New passages can be found in Section 5 around line 620 in the marked-up version of the new manuscript and line around 680 in Section 6.

*Comparison with ACE-FTS*

*In your conclusion you write (L587-589): "Furthermore, we see an excellent comparison with ACE-FTS throughout most of the atmosphere, which suggests the free troposphere and gradient above the UTLS is well represented by TOMCAT$_{OCS}$, and therefore so too are the sources and sinks driving the model. Which shows promise for future OCS work using TOMCAT". I strongly disagree regarding sources and sinks and suggest removing this assertion. It contradicts what you wrote earlier (L351-352): "As ACE measures the upper troposphere and stratosphere primarily, this region is less sensitive to surface processes, so we only compare TOMCAT$_{OCS}$". What you write later (L612-613) is thus more sensible: "While we have shown that TOMCAT$_{OCS}$*

*compares well with satellite observations, the region between the surface and approximately 6 km, which is not measured by ACE-FTS, could hold a lot of information useful in resolving surface fluxes."*

>> "Furthermore, we see an excellent comparison with ACE-FTS throughout most of the atmosphere, which suggests the free troposphere and gradient above the UTLS is well represented by TOMCAT$_{OCS}$, and therefore so too are the sources and sinks driving the model. Which shows promise for future OCS work using TOMCAT." – We agree that parts of this quote required a change and it now reads (line 647):

"Furthermore, we see good comparisons with ACE-FTS throughout most of the atmosphere, which suggests the free troposphere and gradient above the UTLS is well represented by TOMCAT$_{OCS}$. Therefore, there is a suitable balance between model sources and sinks from the surface to simulate atmospheric OCS."

We still assume they are a reasonable set of fluxes, as they are able to model the OCS global spatial distribution in absolute values.

ACE-FTS can test the overall budget in the mid-upper troposphere and give some indication of spatial variations, but it won't discriminate between different processes at the surface. ACE-FTS is an important additional piece of information which can test models, but cannot replace actual surface observations.

**Specific comments**

*L39-40: « This work also shows that the LRU approach is a suitable alternative to mechanistic approaches to quantifying vegetative uptake and will be valuable in using OCS to estimate GPP going forward. » -> It is not clear at all that you made such a demonstration in this study.*

>> Agreed. We have amended this statement to read (line 38): "This work also shows that the LRU approach is an adequate representation of the OCS vegetative uptake, but this method could be improved by various means, such as using a higher resolution GPP product or plant-functional-type-dependent LRU."

*L145-147: Please quantify the uncertainties associated with ACE-FTS retrievals.*

>> The errors in OCS measurements by ACE-FTS are approximately 4% throughout the entire profile, and are approximately 7.2% below 10 km and 3.4% above 20 km. Between these altitudes, the error varies latitudinally and is somewhat dependent on tropopause height but is around 3.8%. An additional sentence has been included at the end of Section 2.1 (line 156):

"The errors in ACE OCS measurements amount to a mean of approximately 3.8% throughout the entire profile globally. In the lower troposphere, below 10 km, errors are larger, approximately 7.2% and above 20 km, are relatively low, at 3.4%."

*L170-173: I understand the COS concentration is computed every 6 hours, but the GPP is available on a monthly basis, please state that the OCS vegetation flux is also computed every 6 hours.*

>> OCS vegetation flux in TOMCAT$_{OCS}$ is indeed calculated every 6 hours. Additional sentence added (line 184): "In the case of TOMCAT$_{OCS}$, the vegetative flux is also calculated every 6 hours."

Additionally, "Each time-step in the model a new F$_{OCS}$ value is calculated" can be found around Line 230. Which was in the second version of the text.

*L204: "Anoxic soil emissions are neglected in this study." -> Justify your choice, their contribution is quantified in Abadie et al. (2022) to help you discuss this matter.*

>> Anoxic soil emissions have undergone several improvements in our understanding and their quantification since the initial model runs in this study. The above phrase has been amended to (line 220): "Anoxic soil emissions are neglected in this study but with the availability of new data sets future simulations could include these sources (Abadie et al. 2022; Whelan et al. 2022)."

*L229-230: "The use of a constant LRU value has been found to contribute less to error in the calculation of Focs than differences in GPP between models on a continental scale (Hilton et al., 2017)." -> Yes, but you are using a single GPP in this study. Plus Maignan et al. (2021) state the opposite (40% of uncertainty on Focs with different land surface models against 70% of uncertainty with different LRUs).*

>> Sentences have been amended to (line 246): "The use of a constant LRU value was found to contribute less to error in the calculation of $F_{OCS}$ than differences in GPP between models on a continental scale by Hilton et al. (2017). However, Maignan et al. (2021) found the opposite, that 70% of uncertainty was attributed to the use of three different LRU datasets and 40% when considering three land surface models. As there are available plant-functional-type-dependent LRU datasets, implementing spatially-varying LRU values will be considered in future work (Seibt et al. 2010; Maignan et al. 2021)."

Ultimately a single LRU was used at the start of the study as no spatially resolved LRU datasets were available at the time, as far as we were aware. However, future TOMCAT studies of OCS can exploit such data sets.

*L235-236: "Only monthly data for 2010 was used, as the interannual variability in the amplitude of the GPP cycle is only about 1% (Chen et al., 2017)." -> As already stated in my former review, the study from Chen et al. (2017) does not conclude that interannual variability in GPP amplitude can be neglected, you have to change your argumentation.*

>> When this work started, we only had a few years of TOMCAT $CO_2$ data to help calculate the vegetation flux. However, it made sense to use the TOMCAT $CO_2$ data since we were using TOMCAT to simulate OCS (e.g. consistent dynamics etc.). As a result, 2010 was chosen as it represented a year approximately in the middle of the study period and would be reasonably representative of the long-term absolute values. Since 2010 was used for the $CO_2$ fields, we then used 2010 for the GPP fields for consistency. The reviewer is correct that in an optimised setup, $CO_2$ and GPP would be annually varying across the full study period. Therefore, in future work on OCS using TOMCAT, longer-term $CO_2$ and GPP data sets can be included.

The following text:

"Only monthly data for 2010 was used, as the interannual variability in the amplitude of the GPP cycle is only about 1% (Chen et al., 2017). Monthly mean gridded $CO_2$ surface mixing ratios for 2010 from a TOMCAT simulation which assimilated surface flask observations of $CO_2$ are used for the $CO_2$ concentration (see Fig. S4 in the supplement) (Gloor et al., 2018). As we compare only monthly means at the surface and seasonal OCS to ACE-FTS, long-term inter-annual variability was not considered in the scope of this work."

Has been amended to (line 254):

"Monthly mean gridded $CO_2$ surface mixing ratios, used to calculate $F_{OCS}$, came from a TOMCAT simulation which assimilated surface flask concentrations for 2010 (see Fig. S4 in the supplement) (Gloor et al., 2018). Given that 2010 is situated approximately in the middle of the study period, it should be a reasonable estimate of the long-term average. Therefore, the GPP data used was also for 2010, given its relatively small inter-annual variability (Chen et al., 2017). As we compare only monthly means at the surface and seasonal OCS to ACE-FTS, long-term inter-annual variability was not considered in the scope of this work. However, future work using TOMCAT can exploit longer-term records of surface $CO_2$ mixing ratios and GPP."

*L242-243: "is slightly under half that of the largest estimation of 1115 Gg S" -> 629 is larger than 557.*

>> under -> over

*L243: "over half that estimated by Launois et al. (2015b), 1335 Gg S" -> 629 is lower than 667.*

>> over -> under

*L246: The title of this section « Balancing the OCS Budget » should be « Manual scaling of the prior fluxes to balance the OCS budget and improve the agreement with the NOAA sites », given lines 277-280 are a bit hidden but justify the notion of « constraint » in your updated title.*

>> The section title has been renamed: 'Scaling of OCS Prior Fluxes to Balance OCS Budget'.

*L323: « which calculates uptake based on leaf surface area, saturation and air pressure » -> The Berry et al. (2013) model uses the atmospheric OCS concentration and a series of conductances.*

>> We have amended the first few sentences of this paragraph having re-read the methodology presented by Berry et al. (2013) (line 347):

"The sink due to vegetation was derived by implementing the OCS vegetative uptake model from Berry et al. (2013) into the land surface model ORCHIDEE, undertaken and explained in detail by Maignan et al. (2021). Berry et al. (2013) calculate OCS uptake using a series of mechanistically and empirically derived conductances that quantify diffusion of OCS from the boundary layer to leaf stomata, where it is eventually hydrolysed by CA in the leaf cell."

*L325: "in the LMDz6 CTM" -> LMDz6 is an Atmospheric Transport Model (no chemistry here).*

>> Done. We have amended this overall sentence to:

"Additionally, Maignan et al. (2021) compare the mechanistic model to the LRU-GPP approach, used in the calculation of OCS in Section 3.3.1, by running the two in the LMDz6 atmospheric transport model."

*L326-327: "it is not adequate for global estimation, unlike the LRU-GPP approach." -> That's a gross misunderstanding, please read what is written in their abstract: "Although the mechanistic approach was more appropriate when comparing to high-temporal-resolution COS flux measurements, both approaches gave similar results when transporting with monthly COS fluxes and evaluating COS concentrations at stations."*

>> We agree. Having re-read Maignan et al. (2021), we have clearly misinterpreted some of their concluding remarks. The sentence has been rephrased (line 352):

"They found that while the mechanistic approach works better on shorter time and smaller spatial scales, both are suitable for global estimation of vegetative OCS uptake".

*L327-330: "Maignan et al. (2021) propose the idea of implementing soil fluxes into the ORCHIDEE land surface model, simultaneously with vegetation, which would be a significant step forward in the capability of constraining and quantifying surface uptake. The reason being: both uptake of OCS by soil and vegetation follows very similar enzymatic pathways, catalysed by carbonic anhydrase (Protoschill-Krebs and Kesselmeier, 1992; Kesselmeier et al., 1999)." -> This is not clear, we need a model for soil exchanges whatever the process, and do not forget about the emission part too, I suggest removing this part.*

>> Done. This is not really in the scope of this work, so we agree removing it is reasonable.

*L332-333: "Note, the vegetation estimate does not match that of Maignan et al. (2021) (-756 Gg S yr-1)." -> As explained in Abadie et al. (2022), they got a different estimate from Maignan et al. (2021) as they considered spatially and temporally varying atmospheric OCS concentrations.*

>> We have removed this sentence and included some additional information in the preceding sentences (line 359):

"Preliminary work on implementing a mechanistic soil uptake model, originating from Ogée et al. (2016), into ORCHIDEE was used as the soil flux in this work (Abadie et al. 2022). Calculation of both vegetation and soil

uptake, using ORCHIDEE, utilise temporally and spatially varying OCS surface mixing ratios (Remaud et al. 2023), obtained from the TM5 atmospheric transport model, driven by posterior fluxes calculated by (Ma et al. 2021)."

*L528: 753 – 756 Gg S yr-1 -> Add the revised vegetation estimate by Abadie et al. (2022).*

>> The estimate by Abadie et al (2022) is now included (576 Gg S) (line 571):

"By utilizing the LRU-GPP approach, we estimate a mean yearly vegetative OCS uptake of 629 Gg S yr-1, which is within the range and uncertainty of the magnitude of this flux from previous top-down studies (see Table 2). Our estimate is also in the range of recent bottom-up estimates by Kooijmans et al. (2021), Maignan et al. (2021) and Abadie et al. (2022) (576 – 756 Gg S yr-1)."

*L571: « by Ogée et al. (2016) » -> Ogée et al. (2016) provides a model, not a distribution of soil fluxes.*

>> We have removed Ogee and left just Abadie et al. (2022).

**Minor comments**

*L53: "This trend is matched stronger positive trends"??*

>> Resolved. "Stronger positive trends are observed in the stratosphere above all sites for 2009-2016, up to 1.93±0.26% yr$^{-1}$"

*L86: "dependent such components as temperature" -> dependent on?*

>> Done.

*L94: suggests -> suggest*

>> Done.

*L186: "our new inventory of fluxes described in Sect. 3.3, TOMCATOCS and to TOMCATSOTA." -> our new inventory of fluxes described in Sect. 3.3, TOMCATOCS and to TOMCATSOTA in Sect 3.4.*

>> Done.

*L230: plant-function-type -> plant-functional-type*

>> Done.

*L278: NOAA-ESRL sites; ALT, BRW and MHD -> NOAA-ESRL sites, ALT, BRW and MHD*

>> Done.

*L283: "to the changes to the vegetative and soil CS2 fluxes."-> OCS fluxes*

>> Done.

*L373: "The fluxes used to model TOMCATOCS reduces" -> reduce*

>> Done.

*L304: "The total oceanic emission has been increased 146%" -> by 146%?*

>> Done.

*L429-430: "The RMSE in Fig. 1 at KUM is reduced by 56.8% and improved SCA compared to TOMCATCON by 81%, from 17.7 ppt to 3.4 ppt. » -> …and the SCA is improved compared…*

>> Done.

L577: *"fair more" -> far more*

>> Done.

L585: *"TOMCATOCS is compared ACE-FTS" -> to ACE-FTS*

>> Done.

L601: *to other work -> to other works*

>> Done.

L883: *Whelan et al. (2018) is not in discussion anymore.*

>> This reference has been updated in all instances, as has the bibliography.

**Reviewer 2**

*The manuscript needs to be read and corrected in detail, also by the co-authors. The revisions are OK, but hardly improved the manuscript, which still adds little to existing knowledge on OCS, except for the comparison to ACE-FTS. I include an annotated pdf with my remarks.*

**Minor comments – from pdf document**

*L48-50: "In the stratosphere the OCS mixing ratio declines strongly with increasing altitude and has a longer mean lifetime than the troposphere, of approximately 64 ± 21 years (Barkley et al., 2008), ranging from 54.1 ± 9.7 years in the sub-tropics to 103.4 ± 18.3 years in the Antarctic (Hannigan et al., 2022)." -> here, this is likely global burden over stratospheric loss. As written now, it sounds like OCS "Lives longer in the stratosphere", which I doubt (that would be calculated as stratospheric burden over stratospheric loss...*

>> Here we are quoting the partial lifetimes in the troposphere and stratosphere which are indeed estimated from the total atmospheric burden divided by the loss in either region. This is done so that the partial lifetimes can be combined to give a total lifetime (i.e. $1/\tau\_total = 1/\tau\_trop + 1/\tau\_strat$). The larger stratospheric lifetime mainly reflects the much lower burden of OCS compared to the troposphere, as well as the higher OH concentrations.

In the revised text we explain the definition of lifetimes quoted. Which can be found around line 53.

*L64: "source" -> sources*

>> Done.

*L86-L87: "dependent such components as temperature, soil moisture, nitrogen content and incident solar radiation" -> ???*

>> Done: "dependent **on** such components as temperature, soil moisture, nitrogen content and incident solar radiation."

*L95: "trend" -> up to 2015?*

>> This trend is up-to and including 2016. Figure 8-11 in Hannigan et al. (2020). "up to 2016".

*L170: "2-3%" -> unclear give OCS loss*

>> We have included values of photolysis: "30 – 34 Gg S yr$^{-1}$ (approximately 3%)"

*L175-176: "varying inter-annual variability" -> varying variabilit?*

>> This has been clarified and made more readable: "Depending on the inventory in use, some vary inter-annually, i.e., vegetative uptake and anthropogenic emission, and the remaining fluxes do not (oceanic emission, soil uptake and biomass burning)."

*L180-182: "Monthly mean surface concentrations are calculated from the flask observations made by the NOAA-ESRL network and compared with monthly mean TOMCAT output averaged across the time period, used for each respective setup." -> would say that co-sampling could be important?*

>> In hindsight, co-sampling the model with the surface observations would offer a more like-for-like comparison. Due to the long life-time and the fact OCS is well-mixed, this should only have a small impact on the study.

We have included the following sentence around line 196: "Co-sampling of the model output with NOAA-ESRL measurements would be a more representative comparison, but here we have not subsampled the model on the specific days of NOAA observations."

*L226: "units of ppb" -> ??*

>> The inclusion of ppb is from a recommendation on the original manuscript: "*Equation 1: Precise the unit for each term of this equation. What is used for OCS background concentration?*", from Referee #1. We have specified parts per billion in the text.

*L227: "typical for TOMCAT" -> ???*

>> Removed this. Unnecessary to include this.

*L227: "We then convert to Gg S yr-1 following simulation." -> ???*

>> Removed this. Again, unnecessary to mention this.

*L238: "concentration" -> concentration or mixing ratio*

>> Mixing ratio. This has been changed here and in other sentences around this highlighted phrase.

*L244-245: "The spatial distribution of $F_{OCS}$ for the months of January, April, July and October, in 2010 only, is presented in the supplement: Fig. S5." -> here I am a bit sceptical, because you can only calculate this flux if you have a full simulation that is providing the OCS mixing ratios. So, the OCS flux can only be given after a simulation that compares well to surface observations. Also, since OCS is changing from year to year, fluxes should also change...*

>> "*here I am a bit sceptical, because you can only calculate this flux if you have a full simulation that is providing the OCS mixing ratios.*" - The vegetative flux is calculated in the first model time-step using an initial value of 500 ppt of OCS. Each successive time-step the flux is calculated using the OCS concentration from the previous one. Therefore a 'full' simulation is not necessary.

"*So, the OCS flux can only be given after a simulation that compares well to surface observations.*" – This is true to an extent. A lot of preliminary testing went into the scaling of fluxes that would yield both a balanced budget and a suitable estimate of the vegetative flux. This is justified in the text by comparing the scaled flux magnitudes to the literature.

"*Also, since OCS is changing from year to year, fluxes should also change...*" – The flux does change. We picked 2010 for simplicity, as the inter-annual variability is not too large. Also, as the photochemistry experiment(s) were executed for 2010 only also.

*L278-279: "the latter of which receives ocean air masses frequently." -> is this referring to MHD? Unclear*

>> This section was removed. While it is true MHD is susceptible to ocean masses frequently, this didn't seem relevant as ocean emissions in the NH were amended as it was the most sensible to change, rather than anthropogenic emissions for example, which has a better certainty.

*L279: "frequently" -> highlighted..*

>> See above.

*Table 1: "-432" -> unclear??? Sinks (1194) and sources (1187) are in balance: something wrong here..*

>> The total source amount in the table was incorrect. This has been amended.

*L355: "stratosphere" -> It would be instructive to include these in the table and to provide a net flux.*

>> As this is a minor test, we believe the text is sufficiently clear at explaining the reduction in photolysis. It is clear that the largest spatial changes in OCS are contained to the stratosphere and do not influence the troposphere hugely in the 1-year test presented.

*L367-368: "is mostly attributed to average concentration." -> ??? unclear*

>> We have changed the structure of the paragraph in Section 4.1 around this comment. And justified the above based on difference in RMSE in Figure 2.

*L403: "to" -> too*

>> Done.

*L505: "The intension if this simulation" -> sorry, but does somebody read the manuscript?*

>> Amended to 'intention of'.

*L528: "Maignan et al., 2021)" -> highlighted*

>> Done. Maignan et al., (2021)

*L530: "CS$_2$ emission" -> from the ocean?*

>> Done. Amended to: "enlarged oceanic CS$_2$ emission source"

*L582-583: "Where TOMCAT$_{CON}$ uses fluxes estimated by Kettle et al. (2002) and TOMCAT$_{SOTA}$ uses a series of novel fluxes from recent literature" -> does not run*

>> We removed the word "where".

*L589: "Which shows promise for future OCS work using TOMCAT." -> is no sentence.*

>> This sentence has been removed.

*L603: "the following changes are necessary in future, such as" -> the following.....such as....bad construct*

>> Improved the sentence structure (line 668): "The following changes are necessary in the future to improve the GPP-LRU approach such as: using inter-annually varying GPP and CO$_2$ mixing ratios, and a temporally and spatially resolved LRU."

*L604: "3" -> i count either 4 or 2....*

>> Changed the sentence to (line 670): "It is challenging to achieve at a high resolution LRU product on a global scale".